# Wnt/PCP controls spreading of Wnt/β-catenin signals by cytonemes in vertebrates

Benjamin Mattes[1,2], Yonglong Dang[2†], Gediminas Greicius[3], Lilian Tamara Kaufmann[4], Benedikt Prunsche[5], Jakob Rosenbauer[6], Johannes Stegmaier[7,8], Ralf Mikut[7], Suat Özbek[9], Gerd Ulrich Nienhaus[2,5,10,11], Alexander Schug[6,12], David M Virshup[3], Steffen Scholpp[1,2]*

[1]Living Systems Institute, School of Biosciences, College of Life and Environmental Sciences, University of Exeter, Exeter, United Kingdom; [2]Institute of Toxicology and Genetics, Karlsruhe Institute of Technology, Karlsruhe, Germany; [3]Program in Cancer and Stem Cell Biology, Duke-NUS Medical School, Singapore, Singapore; [4]Institute of Human Genetics, University Hospital Heidelberg, Heidelberg, Germany; [5]Institute of Applied Physics, Karlsruhe Institute of Technology, Karlsruhe, Germany; [6]John von Neumann Institute for Computing, Jülich Supercomputing Centre, Jülich, Germany; [7]Institute for Automation and Applied Informatics, Karlsruhe Institute of Technology, Karlsruhe, Germany; [8]Institute of Imaging and Computer Vision, RWTH Aachen University, Aachen, Germany; [9]Centre of Organismal Studies, University of Heidelberg, Karlsruhe, Germany; [10]Institute of Nanotechnology, Karlsruhe Institute of Technology, Karlsruhe, Germany; [11]Department of Physics, University of Illinois at Urbana-Champaign, Urbana, United States; [12]Steinbuch Centre for Computing, Karlsruhe Institute of Technology, Karlsruhe, Germany

*For correspondence:
s.scholpp@exeter.ac.uk

Present address: [†]Department of Molecular Neuroscience, German Cancer Research Center, Heidelberg, Germany

**Competing interests:** The authors declare that no competing interests exist.

**Abstract** Signaling filopodia, termed cytonemes, are dynamic actin-based membrane structures that regulate the exchange of signaling molecules and their receptors within tissues. However, how cytoneme formation is regulated remains unclear. Here, we show that Wnt/planar cell polarity (PCP) autocrine signaling controls the emergence of cytonemes, and that cytonemes subsequently control paracrine Wnt/β-catenin signal activation. Upon binding of the Wnt family member Wnt8a, the receptor tyrosine kinase Ror2 becomes activated. Ror2/PCP signaling leads to the induction of cytonemes, which mediate the transport of Wnt8a to neighboring cells. In the Wnt-receiving cells, Wnt8a on cytonemes triggers Wnt/β-catenin-dependent gene transcription and proliferation. We show that cytoneme-based Wnt transport operates in diverse processes, including zebrafish development, murine intestinal crypt and human cancer organoids, demonstrating that Wnt transport by cytonemes and its control via the Ror2 pathway is highly conserved in vertebrates.
DOI: https://doi.org/10.7554/eLife.36953.001

## Introduction

Wnt signaling regulates development and tissue homeostasis in multicellular organisms (*Nusse and Clevers, 2017*), including processes such as cell fate specification, cell proliferation, morphogenesis, and maintenance of tissue integrity. Dysregulation of Wnt/β-catenin signaling has been causally linked to multiple diseases, with Wnt signaling being one of the most frequently dysregulated pathways in several cancer types (*Anastas and Moon, 2013*), including colorectal cancers, pancreatic cancer, and gastric cancers (*Chiurillo, 2015*; *Madan and Virshup, 2015*).

**eLife digest** Communication helps the cells that make up tissues and organs to work together as a team. One way that cells share information with each other as tissues grow and develop is by exchanging signaling proteins. These interact with receptors on the surface of other cells; this causes the cell to change how it behaves.

The Wnt family of signaling proteins orchestrate organ development. Wnt proteins influence which types of cells develop, how fast they divide, and how and when they move. Relatively few cells, or small groups of cells, in developing tissues produce Wnt proteins, while larger groups nearby respond to the signals.

We do not fully understand how Wnt proteins travel between cells, but recent work revealed an unexpected mechanism – cells seem to hand-deliver their messages. Finger-like structures called cytonemes grow out of the cell membrane and carry Wnt proteins to their destination. If the cytonemes do not form properly the target cells do not behave correctly, which can lead to severe tissue malformation.

Mattes et al. have now investigated how cytonemes form using a combination of state-of-the-art genetic and high-resolution imaging techniques. In initial experiments involving zebrafish cells that were grown in the laboratory, Mattes et al. found that the Wnt proteins kick start their own transport; before they travel to their destination, they act on the cells that made them. A Wnt protein called Wnt8a activates the receptor Ror2 on the surface of the signal-producing cell. Ror2 then triggers signals inside the cell that begin the assembly of the cytonemes. The more Ror2 is activated, the more cytonemes the cell makes, and the more Wnt signals it can send out.

This mechanism operates in various tissues: Ror2 also controls the cytoneme transport process in living zebrafish embryos, the mouse intestine and human stomach tumors. This knowledge will help researchers to develop new ways to control Wnt signaling, which could help to produce new treatments for diseases ranging from cancers (for example in the stomach and bowel) to degenerative diseases such as Alzheimer's disease.
DOI: https://doi.org/10.7554/eLife.36953.002

The Wnt signaling network consists of several branches that can be classified according to the receptors involved and the specific signaling cascades that they activate. Two major branches of this network are the β-catenin-dependent pathway and the β-catenin-independent Wnt/planar cell polarity (PCP) pathway (*Niehrs, 2012*). The β-catenin dependent pathway is triggered by the interaction of Wnt with Frizzled (Fzd) receptors and the co-receptor Lrp6 (*Logan and Nusse, 2004*). Wnt/β-catenin signaling regulates the expression of target genes such as *axin2* and *lef1*, as well as the expression of tissue-specific genes, and subsequently controls both cell proliferation and tissue patterning. In the β-catenin-independent Wnt/PCP pathway (*Yang and Mlodzik, 2015*), Wnt proteins bind to Frizzled and to co-receptors such as the receptor-tyrosine kinase-like orphan receptor 2 (Ror2) to regulate cytoskeleton organization by actin polymerization and cell polarity (*Grumolato et al., 2010*; *Ho et al., 2012*; *Oishi et al., 2003*). To this end, the small GTPases Rho, Rac1, and Cdc42 are regulated to control the formation of filopodia and lamellipodia, as well as cell motility and morphogenetic movements of cells in vertebrates. Although the PCP pathway and β-catenin signaling generally act in a mutually repressive fashion, by competing for similar hub proteins such as the effector protein Dishevelled (Dvl) (*van Amerongen and Nusse, 2009*), recent evidence suggests that PCP signaling can act — dependent on the context — either in opposition to, in concert with, or independently of β-catenin signaling.

The production and secretion of Wnt ligands requires lipid modification by the acyltransferase Porcupine (Porcn) followed by binding to Evi/Wls, which serves as a Wnt chaperone and facilitates its transport from the endoplasmic reticulum to the plasma membrane (*Bartscherer and Boutros, 2008*; *Bänziger et al., 2006*; *Yu et al., 2014*). From there, lipophilic Wnt is transported through the neighboring tissue to exert its long-range signaling activity. Extracellular binding proteins have been suggested to increase the solubility of Wg/Wnt in the aqueous extracellular space and facilitate this activity (*Mii et al., 2009*; *Mulligan et al., 2012*). Other studies, however, point to membrane-associated mechanisms of Wg/Wnt delivery, which do not compromise the signaling capability of Wg/

Wnt (*McGough and Vincent, 2016*; *Port and Basler, 2010*; *Stanganello and Scholpp, 2016*). These trafficking routes include Wg/Wnt protein distribution on the plasma membrane of dividing source cells (*Alexandre et al., 2014*; *Farin et al., 2016*) and actively migrating cells (*Serralbo and Marcelle, 2014*), or the dissemination of Wg/Wnt proteins on exovesicles (*Panáková et al., 2005*) (more specifically on exosomes [*Beckett et al., 2013*; *Gross et al., 2012*; *Korkut et al., 2009*]). Wg/Wnt proteins and their receptors are also transported on cell protrusions in various tissues. Lipid-modified Wnt proteins were found at the cell membrane of signaling filopodia — so-called cytonemes — in *Xenopus* and zebrafish (*Holzer et al., 2012*; *Luz et al., 2014*; *Stanganello et al., 2015*), whereas Fzd receptor proteins can be localized to filopodia in *Drosophila* and chicken (*Huang and Kornberg, 2016*; *Sagar et al., 2015*). In zebrafish, an analysis of cytonemes demonstrates that these are specialized filopodia, with stabilizing actin bundles at their cores, which serve as a main transport device for the β-catenin ligand Wnt8a during neural plate patterning (*Stanganello et al., 2015*). Wnt8a is loaded on cytoneme tips and transferred to the neighboring cells by direct cell–cell contact. At the contact sites, Wnt8a cytonemes induce Lrp6/Fzd receptor clustering into the Lrp6 signalosome to activate β-catenin signaling. Although the lengths and numbers of cytonemes are crucial in determining the β-catenin signaling range during embryogenesis (*Stanganello et al., 2015*), it is not yet clear what mechanism controls the formation of Wnt cytonemes in a tissue.

Here, we show that Wnt8a can activate both the PCP pathway by interaction with Ror2 and the β-catenin pathway by interaction with Lrp6. This dual function allows Wnt8a to control its own route of dissemination. In the source cells, Wnt8a binds and activates the Ror2 co-receptor followed by activation of the PCP pathway. Wnt8a-PCP influences convergent extension (CE) movement and activates the small GTPase Cdc42, which leads to the outgrowth of signaling filopodia. Wnt8a is loaded onto these cytonemes, and is transported through the tissue to bind to the β-catenin-specific co-receptor Lrp6 in the responding cells, examples being PAC2 fish fibroblasts and HEK293T human embryonic kidney cells, where it activates the β-catenin pathway in a paracrine fashion. Activation of the β-catenin pathway by Wnt cytonemes leads to target-gene induction, which in zebrafish, regulates neural plate patterning. Ror2-mediated Wnt cytonemes also regulate the proliferation of human gastric cancer cells. We also show that Wnt cytonemes induced by Ror2/PCP signaling are required for the maintenance of murine intestinal crypt organoids. We conclude that Ror2-regulated cytonemes are a critical transport route for Wnt proteins in vertebrates. The molecular mechanism for Wnt cytoneme formation illustrates the co-dependent interactions of the different branches of the Wnt signaling pathways.

## Results

### Tyrosine kinase receptor Ror2 regulates filopodia emergence in vitro

Cumulative evidence indicates that Wnt signal molecules are lipidated and remain associated with membranes during secretion, action and degradation (*Nusse and Clevers, 2017*). Our previous work demonstrated that Wnt molecules can be distributed over distances of 100 µm through a tissue via cytonemes (*Stanganello et al., 2015*). Manipulation of the length or number of Wnt cytonemes led to alterations in Wnt-mediated tissue patterning and malformations of the zebrafish embryo. Therefore, we hypothesized that the formation, emergence, and maintenance of cytonemes are tightly controlled. To identify potential cytoneme regulators, we performed a cell-culture-based genetic screen (*Figure 1A*, *Figure 1—figure supplement 1A–J*) by co-expressing Wnt8a-GFP and glycophosphatidylinositol (GPI)-anchored, membrane-bound mCherry (memCherry) in PAC2 cells together with arrayed cDNA clones from a Medaka cDNA library consisting of 229 kinases (*Chen et al., 2014*; *Souren et al., 2009*). We quantified the length and number of signaling filopodia of ten fibroblasts per cDNA 24 hr post transfection using automated filopodia detection software (*Figure 1B*). The tyrosine-protein kinase transmembrane receptor Ror2 was found to stimulate both filopodia number per cell as well as average filopodia length above the 85th percentile (*Figure 1C, D*).

To validate the screening results, we co-transfected PAC2 fibroblasts with a zebrafish full-length Ror2 expression construct and GPI-memCherry as a membrane marker. The number and length of filopodia were measured in living cells (*Figure 1E*). Ror2 expression significantly increased the average number and length of filopodia per cell (*Figure 1F*, *Figure 1—figure supplement 1A,B*). Ror2

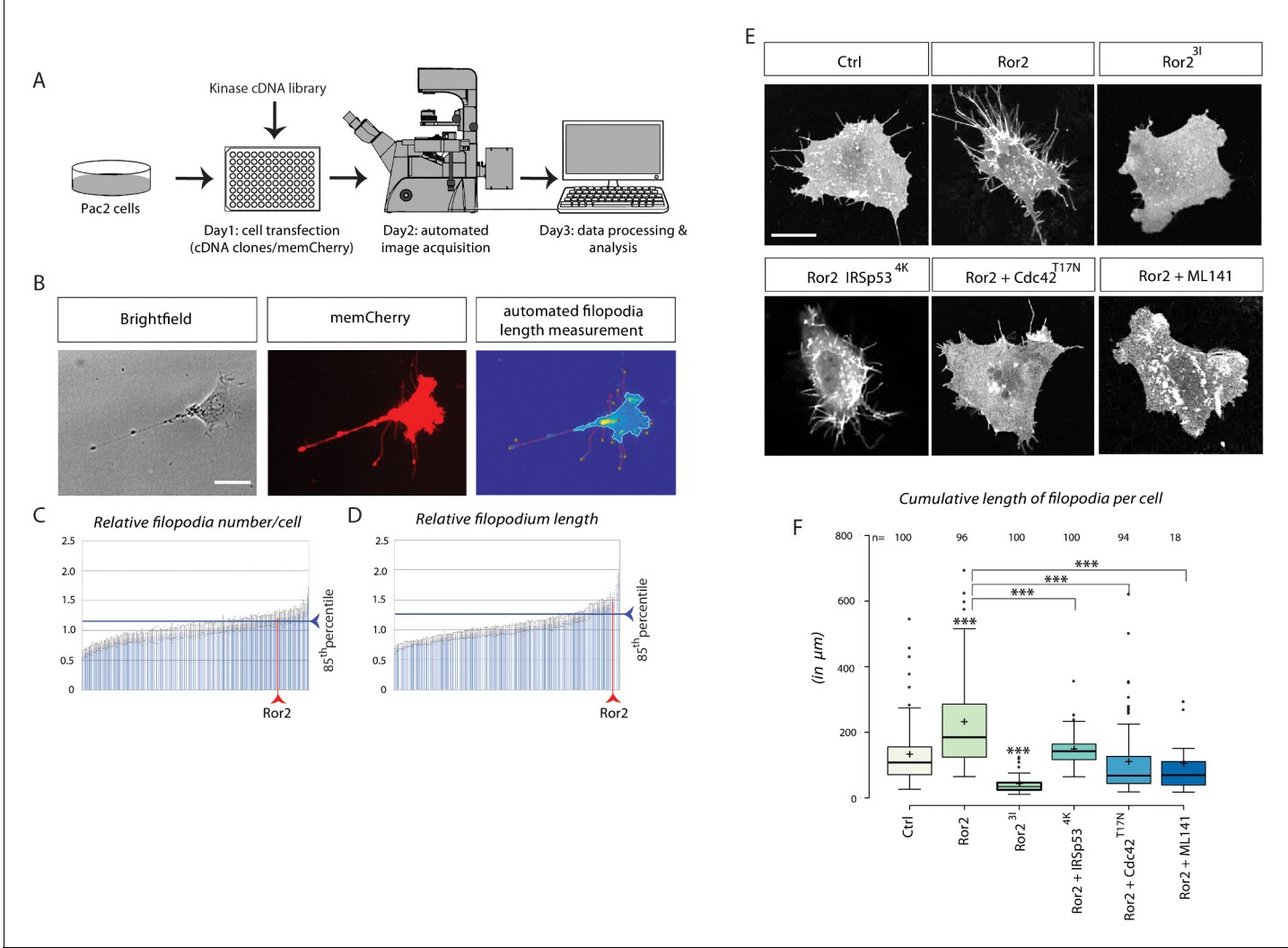

**Figure 1.** Kinase screen and automated image-analysis identifies the receptor tyrosine kinase Ror2 as a potential cytoneme regulator upstream of Cdc42. (**A**) Schematic workflow of the cDNA kinase screen. Wnt8a-GFP and a membrane bound mCherry was co-transfected with kinase library genes in a 96-well plate. Images were acquired automatically and analyzed for filopodia length and numbers using filopodia-detection software. (**B**) Automated image analysis software detects and counts the filopodia of single cells using the memCherry signal and quantifies their length by automatically tracing the tips back to the cell body. Scale bar represents 15 µm. (**C, D**) Diagrams illustrating transfected kinase genes on the x-axes and their relative filopodia number/cell (C) and length (D). Each bar represents one of 229 kinases, sorted ascending by their values. The blue line indicates the 85th percentile (relative number: 1.20, relative length: 1.25). The position of Ror2 in the diagram is highlighted by the red line (relative number: 1.20, relative length: 1.47). (**E**) Quantification of filopodia in PAC2 cells 24 hr after transfection or inhibitor treatment. memCherry was transfected together with the indicated constructs. Scale bar represents 10 µm. (**F**) Boxplot of cumulative filopodia length of cells transfected with the indicated constructs as measured by ImageJ. Centre lines show the median values; box limits indicate the 25th and 75th percentiles as determined by R software; whiskers extend 1.5 times the interquartile range from the 25th and 75th percentiles, outliers are represented by dots. Significance level as indicated: ***=P<0.001, **=P<0.01, =P<0.05. Data meet the assumption that the length of the filopodia is significantly different in the analyzed groups as determined by one-way ANOVA with a $p=5.79*10^{-99}$, confidence interval 95%, F value=118, and df=1,047.

DOI: https://doi.org/10.7554/eLife.36953.003

The following figure supplement is available for figure 1:

**Figure supplement 1.** Quantification of cytoneme parameters.
DOI: https://doi.org/10.7554/eLife.36953.004

requires homodimerization for transautophosphorylation and subsequent downstream signaling (*Liu et al., 2007*), which can be inhibited by overexpressing a kinase-dead construct. Transfection of the dominant-negative mutant Ror2[3I] (*Hikasa et al., 2002*) caused a reduction in both the number of protrusions and their average length, consistent with an essential role of the Ror2 kinase activity

in filopodia induction in PAC2 fibroblasts. The Rho GTPase Cdc42 is crucial for organizing the actin cytoskeleton to stabilize Wnt cytonemes (*Stanganello et al., 2015*) and is thought to be a downstream target of the Wnt/Ror2 pathway regulating filopodia (*Schambony and Wedlich, 2007*). To determine whether Ror2-induced filopodia require Cdc42 function in order to assemble an actin scaffold we co-transfected Ror2-stimulated fibroblasts with dominant-negative Cdc42$^{T17N}$ (*Nalbant et al., 2004*). Blockage of Cdc42 function reduced filopodia formation significantly (*Figure 1E,F*).

BAR-domain containing proteins mold membranes into tube-like filopodia. The insulin receptor tyrosine kinase substrate p53 (IRSp53) is a BAR protein, as well as a Cdc42 effector, which connects filopodia initiation and maintenance by assembling the actin scaffold (*Yeh et al., 1996*). IRSp53$^{4K}$ has four former lysine residues that have been mutated to glutamic acid in the actin-binding sites, inhibiting Cdc42-mediated filopodia formation (*Disanza et al., 2013*; *Kast et al., 2014*). IRSp53$^{4k}$ expression, like Cdc42$^{T17N}$ transfection, blocks Ror2-induced filopodia formation (*Figure 1E,F*). Treatment of Ror2-expressing cells with ML141, a GTPase inhibitor for Cdc42/Rac1 (*Surviladze et al., 2010*), also caused a substantial reduction in both the number and the length of filopodia. Thus, Ror2 is a crucial regulator of filopodia in PAC2 fibroblasts, and filopodia depend on a Cdc42-mediated actin scaffold for their outgrowth and maintenance.

## Cluster formation of Wnt8a and Ror2 is dependent on the CRD domain

Ror2 is a tyrosine kinase receptor that binds Wnt5a via its extracellular cysteine-rich domain (CRD) (*Hikasa et al., 2002*) and serves as a β-catenin independent Wnt co-receptor activating the PCP signaling pathway (*Schambony and Wedlich, 2007*). To investigate the interaction between Ror2 and the β-catenin ligand Wnt8a, we expressed fluorescently tagged Wnt8a and Ror2 proteins in the zebrafish embryo during gastrulation. The mRNA concentration of the injected fluorescent constructs was chosen in such a way that it did not induce phenotypic alteration after 24 hr (*Figure 3—figure supplement 1A*). Zebrafish *ror2* is expressed ubiquitously in early development, with its expression peaking during gastrulation between 2 and 9 hours post fertilization (hpf) (*Bai et al., 2014*). Expression of the β-catenin Wnt ligand *wnt8a* is confined to the embryonic margin during zebrafish gastrulation, orchestrating patterning of the prospective neural plate (*Kelly et al., 1995*; *Rhinn et al., 2005*). Confocal microscopy on live specimens revealed that Wnt8a-GFP displays a punctate pattern in the cytoplasm and at the membrane, including cytoneme tips, but when Wnt is absent, Ror2-mCherry is uniformly distributed in the cell membrane (*Figure 2A*, *Figure 2—figure supplement 1A*). When Wnt8a-GFP and Ror2-mCherry are co-expressed in the same cell, Ror2-mCherry accumulates in punctae along the membrane (*Figure 2A*, *Figure 2—video 1*). Correlated fluorescence intensity analysis of Wnt8a-GFP and Ror2-mCherry suggests that both proteins predominantly co-localize in membrane-associated clusters (*Figure 2B*). This is supported by a Pearson-based correlation analysis of Wnt8a-GFP and Ror2-mCherry in embryonic tissue of a volume of 40 × 40 × 60 μm$^3$ (N=8, Supplementary *Figure 2G*). We found a similar intensity correlation between Wnt5b and Ror2 (*Figure 2—figure supplement 1B,C*). The CRD of Ror2 is crucial for interaction with Wnt ligands (*Hikasa et al., 2002*; *Mikels et al., 2009*). To exclude non-specific clustering of fluorescent fusion proteins, we used a Ror2 construct with a deletion in the Fzd-like CRD. We observed that Ror2-ΔCRD-GFP still localizes to the cell membrane and with Wnt8a-mCherry forming clusters therein. However, image profile analysis showed a marked reduction of the intensity peaks at the cluster site, indicating that Ror2-CRD is required for the interaction with Wnt8a (*Figure 2B*).

## Wnt8a and Ror2 co-migration and protein binding in signaling clusters

We further characterized Wnt8a/Ror2 protein–protein interactions in vivo using line-scanning fluorescence correlation spectroscopy (lsFCS; *Figure 2—figure supplement 1D*), which measures the concentrations and diffusion coefficients of ligands and receptors in the presence of a membrane (*Dörlich et al., 2015*). We performed lsFCS analysis in two different spots, at a Ror2-positive membrane domain (spot 1) and at a Wnt8a/Ror2 membrane cluster (spot 2; *Figure 2C*). A focused laser spot was scanned across the membrane for 390 s while the intensity was measured as a function of time. After compensation for membrane fluctuations, the intensity time traces were time-correlated. In spot 1, we found intensity fluctuation from Ror2-mCherry emission (*Figure 2D*). A fit of the auto-correlation function revealed a receptor concentration (area density) $C_r$=(37 ± 3) μm$^{-2}$. The diffusion

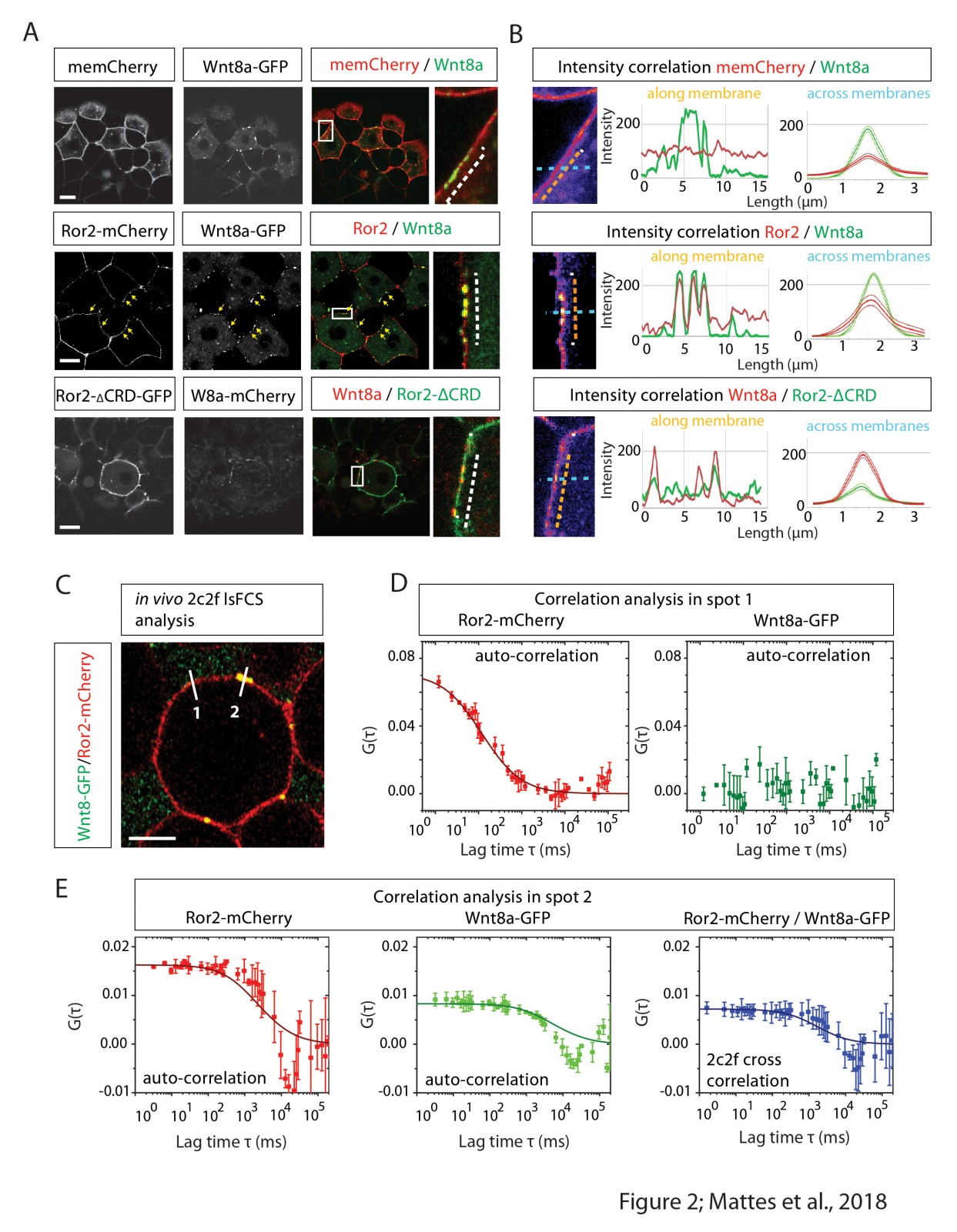

Figure 2; Mattes et al., 2018

**Figure 2.** Image-based protein interaction analysis highlights Ror2s involvement in Wnt8a binding and cluster formation. (**A**) At the 8–16 cell stage, single blastomeres of zebrafish embryos were microinjected with 25 ng/µl mRNA of the indicated constructs to generate cell clones at 50% epiboly for confocal microscopy analysis. The imaging visualizes the co-localization of proteins in a single confocal plane including high-magnification of clusters. (**B**) A co-localization channel, obtained using fire LUT, and the intensity plot profile were generated in ImageJ. The intensity plots show one individual

*Figure 2 continued on next page*

*Figure 2 continued*

measurement along the membrane (left plot, orange arrows) and the average of measurements across the membrane including the SEM (right plot, light blue arrows). The graphs represents 10 membrane clusters acquired from five different cells of two different embryos per treatment. (C) Close-up of a 6–8 hpf live zebrafish embryos showing schematically how two-color two-foci line-scanning fluorescence correlation spectroscopy (2c2f lsFCS) data were collected by laser scanning perpendicular to the membrane (white lines) to collect data in the absence (spot 1) or presence (spot 2) of Wnt8a-GFP. Embryos were mounted in agarose and injected with Wnt8a-GFP and Ror2-mCherry. (D) Autocorrelation functions of Ror2-mCherry (red) and Wnt8a-GFP (green) (symbols) and model fit (line) measured in spot 1. The total time of the measurements was 390 s. Importantly, there is no Wnt8a-GFP at this position, as shown by the lacking autocorrelation in the green channel. Error bars represent standard deviations from two measurements. (E) Autocorrelation functions of Ror2-mCherry (red) and Wnt8a-GFP (green) and the dual-color cross-correlation (blue) (symbols) and model fits (lines) measured in spot 2. Here, Wnt8a-GFP is present, and a high cross-correlation amplitude indicates co-diffusion of bound Ror2-mCherry and Wnt8a-GFP. Error bars indicate standard deviations from two measurements.

DOI: https://doi.org/10.7554/eLife.36953.005

The following video and figure supplement are available for figure 2:

**Figure supplement 1.** Ror2 binds to Wnt8a and Frizzled7a.
DOI: https://doi.org/10.7554/eLife.36953.006

**Figure 2—video 1.** Time-lapse movie of localization of Ror2-mCherry and Wnt8a-GFP in a developing zebrafish embryo in the gastrulation stages.
DOI: https://doi.org/10.7554/eLife.36953.007

coefficient, $D=(0.28 \pm 0.03)$ $\mu m^2$ $s^{-1}$, is similar to values found for LRP6 receptors in the plasma membrane (*Dörlich et al., 2015*). There was no clear emission from Wnt8a-GFP molecules in spot 1. By contrast, lsFCS on spot 2 revealed clear autocorrelations in both color channels (*Figure 2E*), indicating the presence of both Wnt8a-GFP and Ror2-mCherry at this site. We found a dual-color cross-correlation between Wnt8a-GFP and Ror2-mCherry (*Figure 2E*), indicating concerted intensity fluctuations in both color channels, which arise from their co-diffusion in the plasma membrane due to binding. Therefore, the cross-correlation lsFCS data provide clear evidence of complex formation between Wnt8a and Ror2. Furthermore, the low diffusion coefficient of the bound species, $D=(0.02 \pm 0.01)$ $\mu m^2$ $s^{-1}$, indicates that the complexes diffuse as large clusters.

We have previously shown that Fzd7 also interacts with Ror2 and enhances Ror2-mediated signaling during *Xenopus* gastrulation (*Brinkmann et al., 2016*), suggesting that Fzd7a could be a part of the Wnt8a/Ror2 cluster in zebrafish. To test this, we overexpressed Ror2-mCherry, Fzd7a-CFP and Wnt8a-GFP and found co-localization of all three proteins in the cell membrane (*Figure 2—figure supplement 1E,F*).

From our data, we conclude that Wnt8a interacts with Ror2 by binding to its CRD. Wnt8a and Ror2 co-migrate and form dense protein clusters. The data suggest that the Wnt8a/Ror2 clusters are in close steric contact with other components of the Wnt signaling complex including Fzd7a.

## Wnt8a/Ror2 signaling activates the PCP pathway

The interaction of Wnt8a with Ror2 could trigger non-canonical PCP signaling via the Ror2 pathway. PCP signaling plays a role in regulating tissue migration during gastrulation (*Tada and Heisenberg, 2012*). PCP signaling via Ror2 activation regulates collective cell migration towards the embryonic midline, which is most pronounced in the mesodermal germ layer in zebrafish (*Bai et al., 2014*). We utilized this classical PCP-controlled process to observe the involvement of Wnt8/Ror2 in non-canonical signaling. CE can be visualized by condensation of the *no tail* (*ntl*)-positive notochordal plate at the embryonic midline at 11 hpf (*Figure 3A*). To this end, we overexpressed the Ror2 receptor, which alone had a very small effect on the establishment of the *ntl* expression domain (for classification see *Figure 3—figure supplement 1B*). However, overexpression of Wnt8a leads to a broadening and shortening of the *ntl* expression domain. This phenotype is reminiscent of Wnt5b activation. A similar phenotype was observed when Wnt8a and Ror2 were co-expressed. Categorization of the phenotypes suggests that the co-activation of Ror2/Wnt8a and Ror2/Wnt5b have similar effects (*Figure 3B*). Inhibition of Ror2 function by either Ror2[3I] expression or a Morpholino-based Ror2 knock-down also led to CE defects. Our data suggest that Fzd7a may be a member of the Ror2-Wnt8a signaling complex (*Figure 2—figure supplement 1E,F*). We found an enhanced broadening of the embryonic midline if Wnt8/Fzd7a and Wnt8a/Fzd7a/Ror2 were overexpressed (*Figure 3—figure supplement 1C*). This suggests that endogenous Ror2 is already expressed at high levels, and

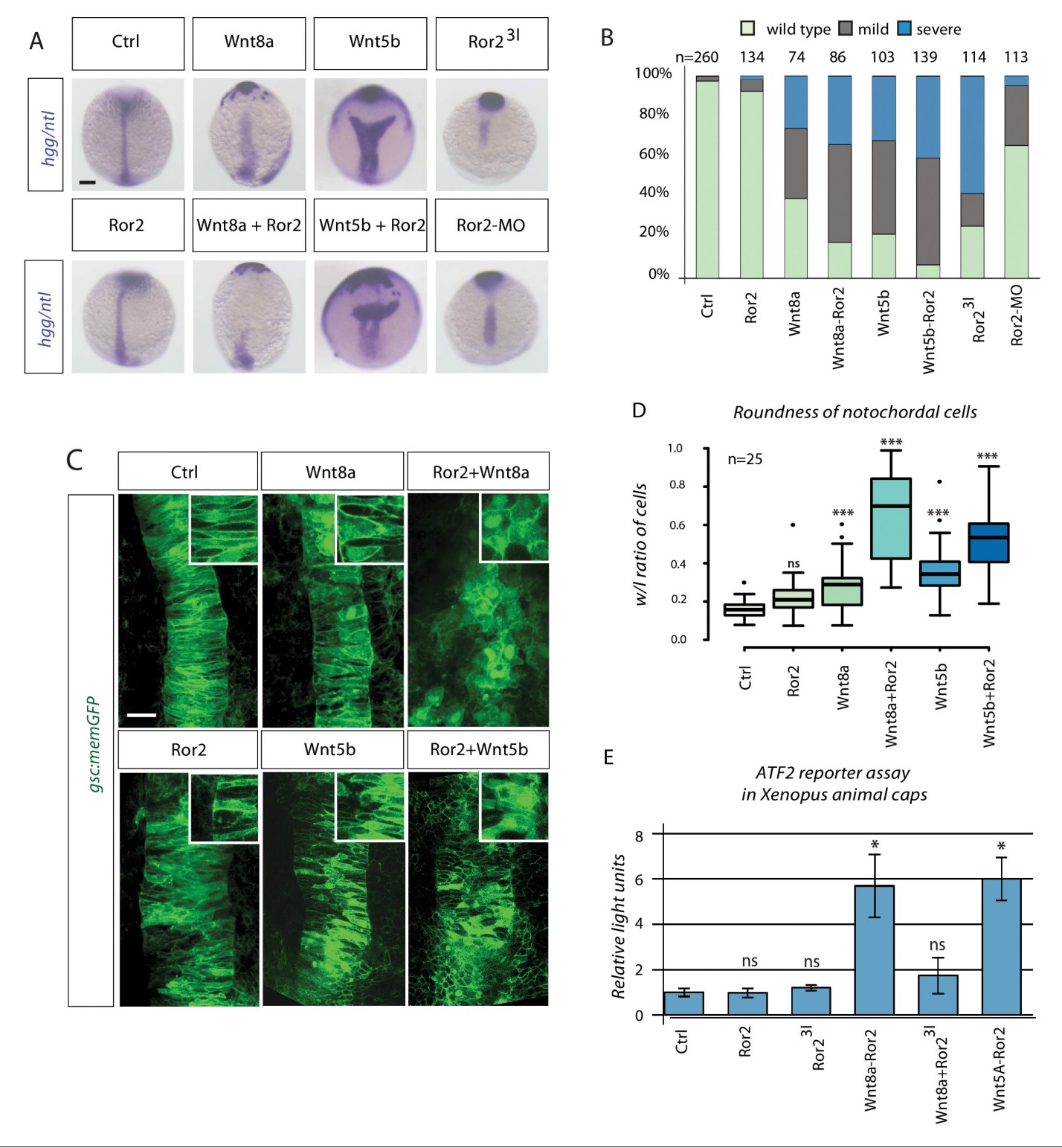

**Figure 3.** Synergistic Wnt PCP activation by Wnt8a and Ror2 in zebrafish and *Xenopus* embryos. (**A**) Embryos at 11 hpf were fixed and subjected to in situ hybridization against *hgg/ntl*. Scale bar represents 200 μm. (**B**) Embryos were sorted into groups according to their phenotypes (see *Figure 3— figure supplement 1* for details). (**C**) *gsc:memGFP* embryos were microinjected with the indicated constructs, fixed at 11 hpf and a defined z-stack was imaged by confocal microscopy. The confocal stack shows the notochord marked by *gsc:memGFP* from a dorsal view with the animal site to the top. Magnified insets highlight the shape of the notochord cells. Scale bar represents 20 μm. (**D**) Analysis of cell roundness. The boxplot shows the width/ length ratio of 25 notochordal cells. Circularity ranges from 0 (infinitely elongated polygon) to 1 (perfect circle). ANOVA confirmed that cell roundness is

*Figure 3 continued on next page*

*Figure 3 continued*

significantly different in the analyzed groups with p=2.91*10$^{-24}$, confidence interval 95%, F value=37, and df=149. (E) ATF luciferase reporter assay of pooled *Xenopus* gastrulae injected with the indicated constructs and the ATF2 firefly and Renilla luciferase reporter. Data shown are the mean with SD of three independent experiments. \*\*\*=P<0.001, \*\*=P<0.01, \*=P<0.05, ns, not significant.

DOI: https://doi.org/10.7554/eLife.36953.008

The following figure supplement is available for figure 3:

**Figure supplement 1.** Analysis of convergent extension phenotype in zebrafish development

DOI: https://doi.org/10.7554/eLife.36953.009

that the available Wnt ligand concentration is the key quantity-controlling step for PCP signaling during zebrafish CE.

During CE, cells intercalate in the notochordal plate (convergence), push previously adjacent cells apart, and lengthen the field along the AP axis (extension) (*Glickman et al., 2003*). We investigated the shape of the notochord cells in embryos with ectopically expressing Wnt8/Ror2 signaling. We found that the cells had a less bipolar shape and displayed a more circular form in embryos with Wnt8/Ror2 signaling, reminiscent of Ror2 activation by Wnt5a (*Figure 3C,D*), suggesting that medio-lateral narrowing of axial mesoderm is reduced.

Activation and inhibition of the PCP signaling pathway leads to a similar phenotype (*Tada and Heisenberg, 2012*). Therefore, we were unable to distinguish how Wnt8a/Ror2 alters PCP signaling. During *Xenopus* gastrulation, Wnt5A activates Ror2 downstream signaling, leading to Cdc42 activation, JNK phosphorylation, and ultimately, the enhancement of ATF2 transcription (*Hikasa et al., 2002*; *Schambony and Wedlich, 2007*). To test whether zebrafish Wnt8a is able to activate Ror2 signaling, we used a reporter assay with an ATF2-responsive element driving luciferase expression in *Xenopus* embryos (*Brinkmann et al., 2016*; *Ohkawara and Niehrs, 2011*). Wnt8a co-expressed with Ror2 produced a greater than five-fold induction of the ATF2 reporter in *Xenopus* (*Figure 3E*). Co-expression of Wnt5A/Ror2 leads to a similar activation of reporter expression, whereas expressed Ror2 without a co-expressed ligand did not alter expression of the ATF2 reporter. We determined whether the kinase domain of Ror2 is required for Wnt8a-dependent activation of the PCP pathway by overexpressing Wnt8a together with dominant-negative Ror2[3I] and observed a reduction of ATF2 reporter activation but an activation of Wnt8a/Ror2.

Taken together, our data indicate that Wnt8a serves as a ligand for the receptor Ror2 and induces PCP signaling upon binding. Thus, ectopic overexpression of Wnt8a modulates cell movements and cell morphology in zebrafish and gene transcription in *Xenopus*.

## Ror2 induces Wnt-cytonemes during zebrafish neural patterning

To study the dynamics of filopodia formation in the Wnt8a-positive germ ring during normal development in zebrafish (*Figure 4—video 1*), we quantified signaling filopodia during gastrulation in live embryos. In confocal image stacks, filopodia were traced using a semi-automatic live wire approach (*Barrett and Mortensen, 1997*). A precise measurement of filopodia protrusion lengths in 3D was obtained using manually selected nucleation start points and filopodia end points. We found that the number of filopodia significantly increase from in the period from 4 hpf to 6 hpf, which comprises the neural-plate-patterning phase (*Figure 4A,B*). This coincides with increasing Ror2 expression levels during zebrafish development (*Bai et al., 2014*). We determined whether the formation of these filopodia was dependent on Ror2 function by manipulating Ror2 signaling and measured germ ring cell filopodia number at 6 hpf. We found only a modest increase in filopodia number when Ror2 was activated, suggesting that Ror2 itself is not rate-limiting (*Figure 4C,D*). However, when Ror2 function was reduced by overexpression of Ror2[3I], we observed a significant reduction in filopodia number. We conclude that Ror2 signaling is required for filopodia induction of embryonic marginal cells during zebrafish development.

We speculated that Ror2 might influence the formation of filopodia carrying Wnt8a protein (Wnt8a-cytonemes) during zebrafish gastrulation. To visualize these cytonemes, we generated cell clones at the embryonic margin expressing Wnt8a-GFP and memCherry. Wnt8a-GFP clusters were seen in the cell membrane and the cytoneme tips of germ ring cells (*Figure 4E*). Statistically, more filopodia carrying Wnt8a-GFP clusters on their tips were detected upon Ror2 overexpression within

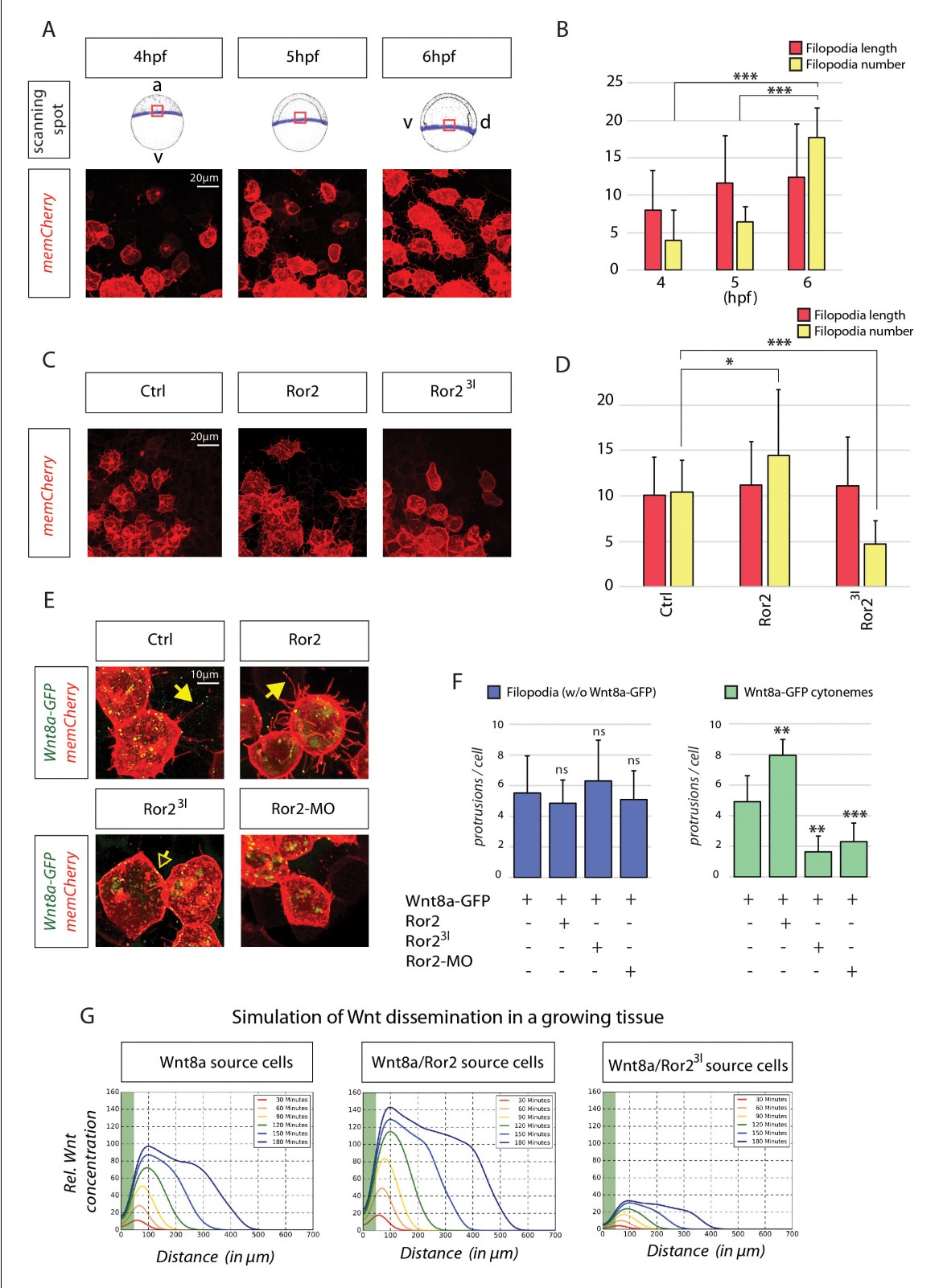

**Figure 4.** In-depth analysis and simulation of Wnt-positive cytonemes in zebrafish embryos in vivo. (**A**) Live confocal microscopy analysis of filopodia dynamics over time. Mosaic expression of memCherry was utilized to quantify the protrusions of single cells at the indicated positions. The image shows the same cells at different times during zebrafish gastrulation. (**B**) Filopodia of cells in (A) were measured using semi-quantitative segmentation software (see *Figure 4—figure supplement 1* for details). Quantification illustrates the mean filopodia length and the number of filopodia per cell with

*Figure 4 continued on next page*

*Figure 4 continued*

SEM at different time points. (**C**) Effect of Ror2 on filopodia length and number. Live confocal microscopy analysis of the filopodia of embryos injected with Ror2 or Ror2[3l] mRNA at the one-cell stage. (**D**) Diagram of mean filopodia length and number per cell with SEM. (**E**) Analysis of Wnt cytonemes during live imaging. Embryos were microinjected at the 16-cell stage to generate a cell clone expressing Wnt8a-GFP and memCherry to visualize cytonemes and the Ror2, Ror2[3l] or Ror2-MO oligomer. Confocal images of single cells were taken and subjected to filopodia length/number measurement. See ***Figure 4—figure supplement 1*** for stacked diagrams with chi-squared test analysis. (**F**) Bar diagram showing the number of filopodia without Wnt8a-GFP (i.e. GFP signal below detection limit) or filopodia carrying Wnt8a-GFP (cytonemes) (i.e. GFP signal above detection limit) per cell. Statistical analysis \*\*\*=P<0.001, \*\*=P<0.01, \*P<0.05. Data meet the assumptions illustrated in (**B**) that filopodia length and number are significantly different at the analyzed time-points (One-way ANOVA, p=1.35\*10$^{-9}$, confidence interval 95%, F value=27.6, and df=73) and in (**D**) that the filopodia number and length are significantly different across treatments (p=1.61542\*10$^{-7}$, confidence interval 95%, F value=24.6, and df=39). (**G**) Simulation of Wnt dissemination in a growing tissue over 180 min. Wnt8a is distributed in a graded manner in the target tissue. Modeling the dynamically expanding tissue with single-cell resolution and discrete implementation of cytoneme-based transport establishes a morphogen gradient over the length of the developing tissue. On the basis of the cytoneme appearance measured in (**D**) and (**F**), Ror2 activity in the Wnt-producing cells alters the input of Wnt in the target field, which is directly incorporated into the simulation by altering the formation frequency of cytonemes and thus ligand concentration in the neighboring tissue. Simulated time in minutes after the onset of Wnt production at 4 hpf.

DOI: https://doi.org/10.7554/eLife.36953.010

The following video and figure supplement are available for figure 4:

**Figure supplement 1.** Cytoneme quantification software toextract filopodia automatically using the memCherry channel of the acquired images.

DOI: https://doi.org/10.7554/eLife.36953.011

**Figure 4—video 1.** Time-lapse movie of clonal cells forming cytonemes in a developing zebrafish embryo during gastrulation.

DOI: https://doi.org/10.7554/eLife.36953.012

cells (***Figure 4E,F***). Conversely, we found a significant reduction in the number of cytonemes in Ror2-deficient marginal cells. Wnt8a-GFP is still present at the plasma membranes of cells that have compromised Ror2 function, suggesting that intracellular routing of Wnt8a from the producing organelles to the cell membrane is independent of Ror2-dependent cytonemal transport. Filopodia without detectable Wnt8a seemed to be unaffected by Ror2 signaling in zebrafish (***Figure 4F***). This suggests a function of Ror2 in regulating a specific subset of filopodia, those carrying Wnt8a (the Wnt cytonemes), in the zebrafish embryo in vivo.

On the basis of these findings, we hypothesized that Ror2 function in the Wnt source cells is crucial for Wnt dissemination via cytonemes. To test the consequences of altered Ror2 signaling quantitatively, we used a simulation of morphogen distribution via cytonemes (***Stanganello et al., 2015***). The simulation takes into account ligand transport by cytonemes, ligand decay and the migration of epiblast cells using a Monte-Carlo-based direct event simulation approach. Employing cytonemes as the exclusive transport mechanism from the producing cell group to the target cell group, and assuming an unlimited source of the ligand, we found that cytonemes can distribute Wnt8a in a graded manner in the dynamically evolving target tissue (***Figure 4G***). We tested two scenarios with varying cytoneme number, based on our in vivo measurements after alteration of Ror2 function (***Figure 4F***). We found that ligand concentration within the morphogenetic field depends on the appearance of the cytonemes (***Stanganello et al., 2015***). We also found that increasing the number of cytonemes per cell (by the experimentally determined factor of 1.61) after expression of Ror2 and Wnt8a in the source cells leads to an enhanced ejection (187%) of the ligand into the target tissue when compared to that in the Wnt8a-producing cells (***Figure 4G***). Consistently, by using the measured cytoneme parameters after blockage of Ror2 function (number scaled by 0.388), we found a decrease to 36% of the Wnt8a concentration compared to the WT situation and a smaller range of the morphogen gradient within the tissue. These data suggest that Ror2 signaling can specifically regulate the number of Wnt-positive cytonemes in vivo and thus represents a cytoneme-specific regulator. Furthermore, on the basis of the simulations, we predicted a strong decrease in the range of Wnt signal activation in the neighboring tissue when Ror2 function is compromised in the Wnt8a source cells.

## Ror2 presents Wnt8a to the target cell to induce a ligand-receptor cluster

Next, we asked how Wnt8a and Ror2 interact to facilitate cytoneme-mediated transport. We used a high-resolution imaging approach in the developing zebrafish embryo that involved the

overexpression of tagged constructs (*Figure 5*). The high-sensitivity of this image-based approach allowed us to reduce the expression levels of the tagged construct significantly, and morphological alterations of the embryonic phenotype were not observed at 24 hpf (*Figure 3—figure supplement 1A*). Using a time-lapse analysis, we observed the formation of Wnt8a-GFP-positive clusters, which transit in the plasma membrane of the secreting cell (*Figure 5B*). Then, Wnt8a co-localizes with Ror2 at the plasma membrane, suggesting Wnt8a-Ror2 cluster induction (*Figure 5C*) as previously described (*Figure 2A,B*). These clusters initiate cytoneme formation and, consequently, they decorate the tip of the outgrowing cytoneme. The cytoneme contacts the target cell and Wnt8a-Ror2 forms a cluster at the receiving cell (*Figure 5C*, *Figure 5—video 1*). Within minutes, this cluster is endocytosed into the target cell. We wondered whether Wnt8a-Ror2 induces the Wnt signaling cascade in the target cell. Therefore, we analyzed Lrp6-signalosome formation at the plasma membrane of the target cell by induction of a Wnt8a-secreting cell clone and a Lrp6-expressing receiving cell clone (*Figure 5D*). We find that cytonemal Wnt8a-mCherry induces Lrp6-GFP cluster formation at the membrane of the target (*Figure 5E,F*). We hypothesized that the source cell presents Wnt8a by clustering the ligand on Ror2-positive cytonemes. Indeed, we observe a Lrp6-GFP cluster at the contact points of Ror2-positive cytonemes (*Figure 5E*). The following dynamics of the Lrp6-signalosome have been described recently in zebrafish development (*Hagemann et al., 2014*).

Therefore, we conclude that Ror2 clusters on cytoneme tips to act as a platform to present Wnt8a to the target cell and to induce the Wnt signaling cascade therein.

## Ror2 regulates PCP signaling in the Wnt source cells and β-catenin signaling in the Wnt receiving cells

On the basis of our simulations, we speculated that Ror2 signaling may have a function in Wnt ligand trafficking and, consequently, in paracrine β-catenin signaling during zebrafish gastrulation. To test this, we overexpressed Ror2 and analyzed the effect of this overexpression on CE processes (via PCP signaling) and, simultaneously, on neural plate patterning (via β-catenin signaling; *Figure 6—figure supplement 1A*) during embryogenesis. Overexpression of Ror2 by injection of low levels of mRNA did not induce gross morphological changes in zebrafish embryos, consistent with our findings that Ror2 without a suitable ligand only mildly impacts PCP-mediated processes (*Figure 3A,C, E*). Overexpression of Wnt8a resulted in a substantial alteration in neural plate patterning as described above, with β-catenin signaling being activated in the entire neural plate, marked by ubiquitous *axin2* expression at 6 hpf (*Figure 6A*). As a consequence, we observed posteriorization of the developing nervous system, observed as an anterior shift of the *fgf8a*-positive midbrain-hindbrain boundary (MHB) at 9 hpf and a loss of the anterior *pax6a*-positive forebrain at 26 hpf. In embryos co-expressing Wnt8a together with Ror2, we still observed the posteriorization phenotype in the neural plate and, in addition, we found that CE is compromised, as the neural plate does not converge to the midline and, consequently, the expression domains of *fgf8a* at the MHB showed a pronounced gap.

We compared these observations to embryos expressing the β-catenin-independent ligand Wnt5a, and Wnt5a together with Ror2. Ror2-mediated Wnt5a signaling induces CE in *Xenopus* (*Hikasa et al., 2002*) and represses β-catenin signaling in mouse embryos (*Mikels et al., 2009*). In both settings, we observed a strong effect on CE movement in the zebrafish embryo. In addition, Wnt5a/Ror2 overexpression led to reduced β-catenin signaling, causing a reduction in target gene expression (*axin2*) and anteriorization of the neural plate, leading to a posterior shift of *pax6a* expression in the forebrain. We conclude that Wnt8a can activate β-catenin signaling and PCP signaling via the Ror2 receptor during zebrafish development. We showed that Wnt5b/Ror2-mediated PCP signaling represses Wnt/β-catenin signaling. We hypothesized that Wnt8a function depends on the route of secretion. However, global overexpression did not differentiate between autocrine and paracrine Wnt8a signaling mechanisms and on subsequent downstream activation.

To separate Wnt-producing from Wnt-receiving cells, we performed a co-cultivation assay using HEK293T cells, which are typically Wnt-Off due to low endogenous expression of Wnt ligands (*Voloshanenko et al., 2017*). Cytoneme regulators were transfected into HEK293T source cells (Wnt-producing cells) and co-cultivated with HEK293T cells expressing the SuperTOPFlash TCF/Wnt reporter, with seven TCF-responsive elements hooked up to nuclear mCherry (7xTRE-NLS-mCherry [*Moro et al., 2012*], *Figure 6—figure supplement 1B*). Ror2 transfection into source cells did not alter the induction of 7xTRE-nucRFP in the receiving cells (*Figure 6B,C*). However, Wnt8a-producing

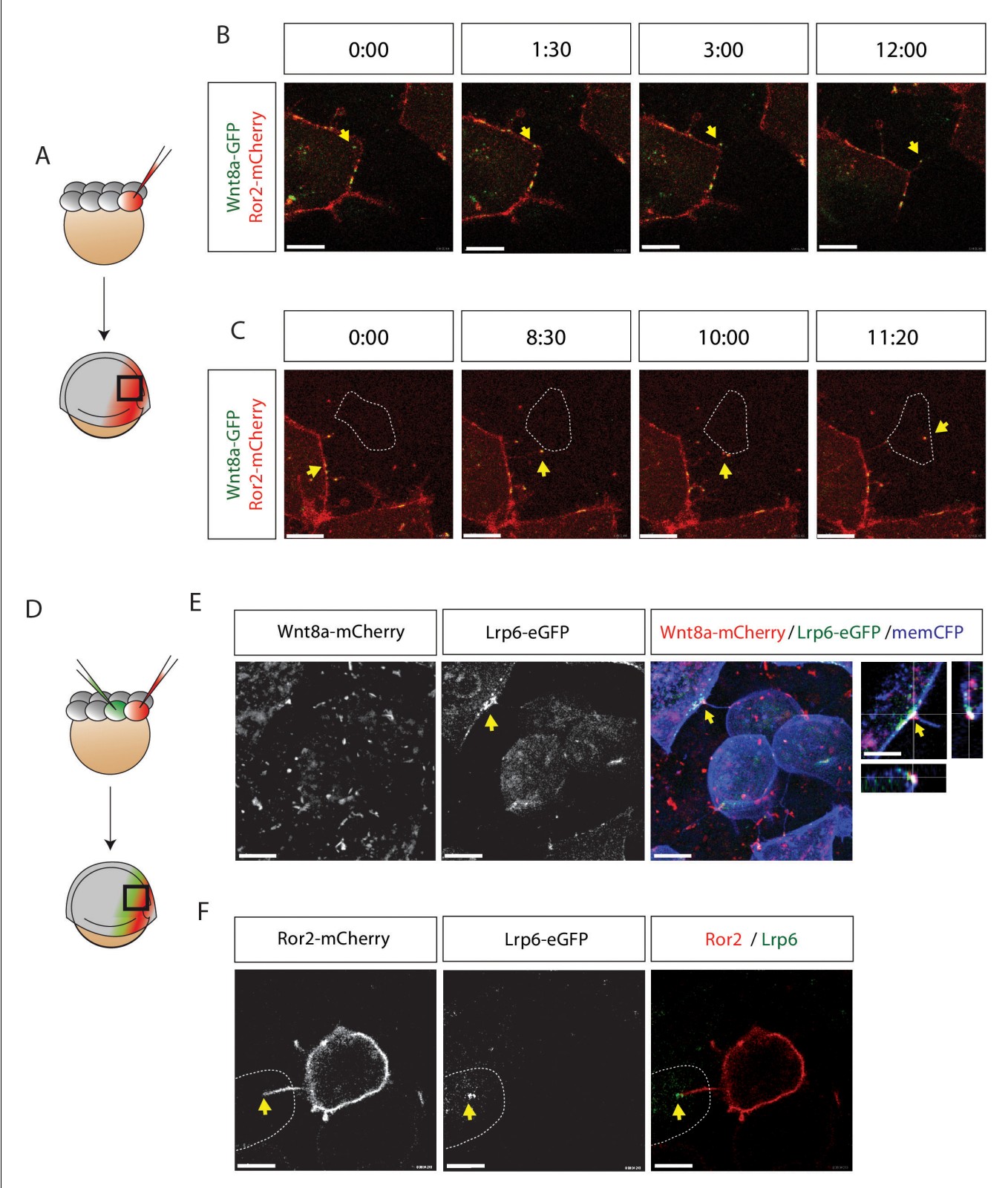

**Figure 5.** Visualization of Ror2-Wnt8a cytonemal transport and Lrp6 receptor clustering in the living zebrafish. (A) Illustration of the injection procedure to generate single cell clones. (B–C) Time series of a single confocal plane of Ror2-mCherry/Wnt8a-GFP-expressing cells to (B) observe Wnt8a recruitment to the membrane and cytoneme initiation and (C) cytonemal target finding and Ror2/Wnt8a cluster endocytosis into the receiving cell. Optimal microscopical settings were used to allow an improved axial resolution of ca. 250 nm. Single channels in *Figure 5—figure supplement 1A,B*

*Figure 5 continued on next page*

*Figure 5 continued*

and *Figure 5—video 1*. Yellow arrows indicate the Wnt8a/Ror2 cluster and highlights pruning of the cytoneme tip after successful cytonemal delivery. Yellow arrows mark the Wnt8a/Ror2 cluster. (**D–F**) Wnt8a-mCherry cytoneme leads to Lrp6-GFP accumulation and co-localization at the receiving cell. (**D**) Illustration of the injection procedure used to generate clones to visualize paracrine signal transport. (**E**) 3D-stack of a Wnt8a-mCherry cytoneme leading to Lrp6-GFP accumulation and co-localization at the receiving cell. The close-up shows a single plane including orthogonal YZ and XZ views of the cytoneme contact point. (**F**) Single-plane image of a Ror2-mCherry/Wnt8a cell leading to Lrp6-GFP clustering on the cytoneme contact site. Scale bars = 10 µm; in E = 20 µm.

DOI: https://doi.org/10.7554/eLife.36953.013

The following video and figure supplement are available for figure 5:

**Figure supplement 1.** Visualization of Ror2-Wnt8a cytonemal transport in the living zebrafish.
DOI: https://doi.org/10.7554/eLife.36953.014
**Figure 5—video 1.** Time-lapse movie of clonal cells forming cytonemes in a developing zebrafish embryo during gastrulation.
DOI: https://doi.org/10.7554/eLife.36953.015

cells led to the activation of signaling activity in the HEK293T reporter cells. Reporter expression was further enhanced (147.3% compared to Wnt8a transfected source cells) when Wnt-producing cells co-expressed Wnt8a and Ror2, indicating a synergistic interaction between Wnt8a and Ror2 (*Figure 6D*). Co-transfection of Wnt8a with dominant-negative Ror2[3I] resulted in a 34.6% decrease in reporter activation below the level seen in Wnt8a-transfected source cells. These findings support our simulations regarding the available Wnt8a concentration in the target tissue after alteration of Ror2-dependent cytoneme appearance (*Figure 4G*). We conclude that Wnt8a can be transmitted via Ror2-dependent cytonemes in HEK293T cells, whereas Wnt5b and Wnt5b/Ror2 transfections were unable to activate the β-catenin signaling reporter in neighboring cells (*Figure 6D*).

To test whether Ror2-mediated Wnt cytonemes affect β-catenin-dependent target gene activation in neighboring cells in vivo, we generated small-source clones by microinjecting cytoneme regulator mRNAs at the 16-cell stage (*Figure 6—figure supplement 1C*). By mid-gastrulation, the source cells were distributed over an area of the embryo and intermingled with WT host cells, generating many responding cells around a few source cells. At 6 hpf, we analyzed the transcriptional profile of the embryos for the β-catenin target genes *axin2* and *lef1*. Embryos with cells overexpressing Ror2 or Ror2[3I] showed no alteration in *axin2* or *lef1* expression (*Figure 6D*). However, source cells overexpressing Wnt8a resulted in a significant increase in β-catenin-dependent target gene expression, which was not further enhanced by Ror2 addition. However, blockage of cytoneme formation in the Wnt8a source cells by co-expression of Ror2[3I] led to a significant reduction of β-catenin target gene induction in neighboring cells. Blockage of filopodia per se by overexpression of the dominant-negative form of IRSp53[4K] caused a similar reduction of activation of *axin2* and *lef1* expression in embryos. This suggests that, during zebrafish gastrulation, the majority of Wnt8a protein is transmitted via cytonemes and that the formation of these Wnt cytonemes is Ror2 dependent.

## Ror2-dependent cytonemes in cancer cell lines and intestinal organoids

Over-activation of canonical β-catenin signaling can be identified in one-third of gastric cancers (*Chiurillo, 2015*). β-catenin signaling is essential for self-renewal of gastric cancer stem cells, leading to Wnt-mediated resistance to apoptosis, which may be responsible for recurrences of these tumors. The β-catenin-independent branch plays a similarly important role in cancer progression: the key ligands Wnt5a and Ror2 are upregulated in various gastric cancers, regardless of the histological phenotype. To determine whether Wnt ligands are transported on cytonemes between gastric cancer cells, we used the gastric cancer (GC) cells lines MKN7, MKN28 and AGS. Transfected Wnt8a-mCherry localized on filopodia in GC cell lines (*Figure 7A*, *Figure 7—figure supplement 1*). Forced expression of Ror2 led to a strong increase in the number of filopodia on GC cells (*Figure 7B,C*), whereas there was a significant reduction in cumulative filopodia length in cells expressing dominant-negative Ror2[3I] (*Figure 7C*), indicating that Ror2 also control filopodia formation in GC cells.

We focused on AGS cells because they show highly dynamic formation and retraction of filopodia, are particularly receptive to Ror2 manipulation, express Wnt1 at constant high levels, and, thus, have high endogenous β-catenin activity, which has been linked to the increased proliferation rate of this cell line (*Mao et al., 2014*). To assess whether cytoneme-mediated Wnt transport influences AGS cell behavior, and specifically proliferation, we co-cultivated Ror2-transfected cells with cells

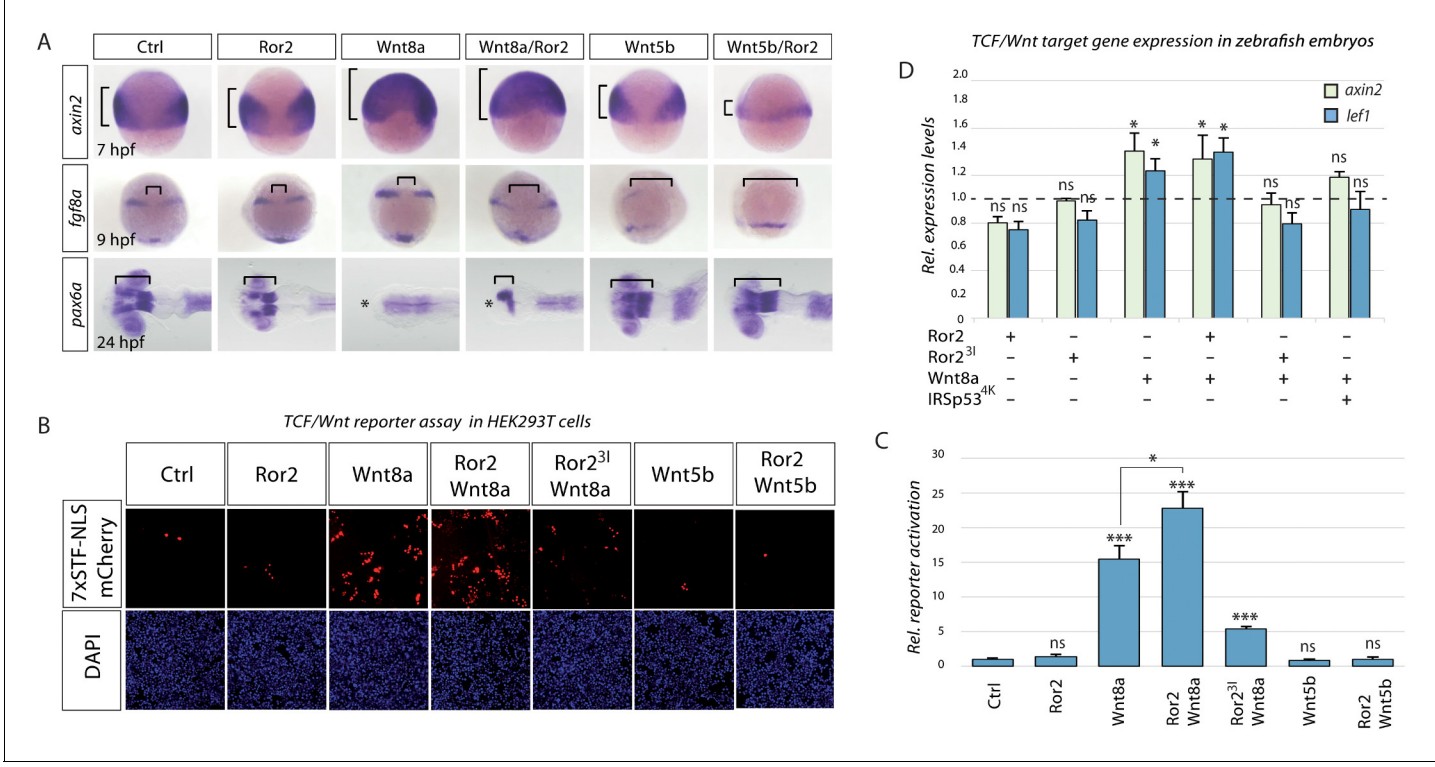

**Figure 6.** Ror2 enhances paracrine Wnt/β-catenin signaling in zebrafish embryos and HEK293T co-culture. (A) Microinjected embryos at the indicated stages were fixed and subjected to in situ hybridization against the Wnt target gene (*axin2*) or against markers for brain patterning (*fgf8a*, *pax6a*). Brackets indicate the expansion of Wnt target gene expression (*axin2*), distance of fgf8 expression domains or forebrain territory (*pax6a*), while asterisks indicates a lack of forebrain tissue. (B) Co-culture reporter gene assay in HEK293T cells. The TCF/Wnt reporter population was transfected with 7xTRE Super TOPFlash-NLS-mCherry and Lrp6 and co-cultured with a second population as indicated. The co-culture principle is illustrated in *Figure 6—figure supplement 1*. (C) Relative reporter activation by measuring the mCherry signal. The bar chart shows the mean value with SEM of three independent experiments. Scale bar represents 200 μm. (D) RT-qPCR-based expression assay in zebrafish embryos. Embryos were injected at the 16-cell stage to generate a distinct signaling center at 75% epiboly (see *Figure 6—figure supplement 1* for details) and were subjected to RTqPCR. Relative ΔΔCt expression values of Wnt-β-catenin target genes *axin2* and *lef1* are acquired by normalization to actb1 and in comparison to a control-injected sample (relative ctrl expression level shown as a dotted line). Each condition represents the mean Rt-qPCR result of 4 × 50 embryos acquired from two independent experiments. *** P < 0.001, * P < 0.05. ANOVA confirmed the hypothesis that expression levels are different between treatments with a p=0.00025, confidence interval 95%, F value=33.9, df=11.

DOI: https://doi.org/10.7554/eLife.36953.016

The following figure supplement is available for figure 6:

**Figure supplement 1.** Methodolocical illustration for AP patterning and co-cultivation assay.

DOI: https://doi.org/10.7554/eLife.36953.017

carrying the nuclear marker nucRFP. Cells overexpressing Ror2 significantly increase cell proliferation in neighboring AGS cells (*Figure 7D,E*). Coexpression of the specific filopodia inhibitor IRSp53[4K] dampened the increased proliferation rate induced by Ror2 expression. Inhibition of Wnt signaling by the tankyrase inhibitor IWR1 abrogated the stimulatory effect of Ror2 expression, confirming that the Ror2 effect is due to increased Wnt signaling. We conclude that Wnt is moved on cytonemes between GC cells to stimulate Wnt/β-catenin signaling and proliferation of neighboring cells. Abrogation of this transport route has a consequence similar to that resulting from inhibition of Wnt signaling per se – it leads to reduced proliferation.

As the intestinal crypt requires a constant supply of Wnt signaling for tissue homeostasis (*Beumer and Clevers, 2016*; *Kuhnert et al., 2004*; *Pinto et al., 2003*; *Sailaja et al., 2016*), we asked whether Wnt cytonemes operate in the mouse intestinal crypt. In vivo, subepithelial myofibroblasts provide the major source of physiologically relevant Wnts, which maintain the crypt in vivo (*Kabiri et al., 2014*; *Valenta et al., 2016*). It has been further demonstrated that these PdgfRα-positive myofibroblasts regulate the intestinal stem-cell niche by Wnts and RSPO3 (*Greicius et al.,*

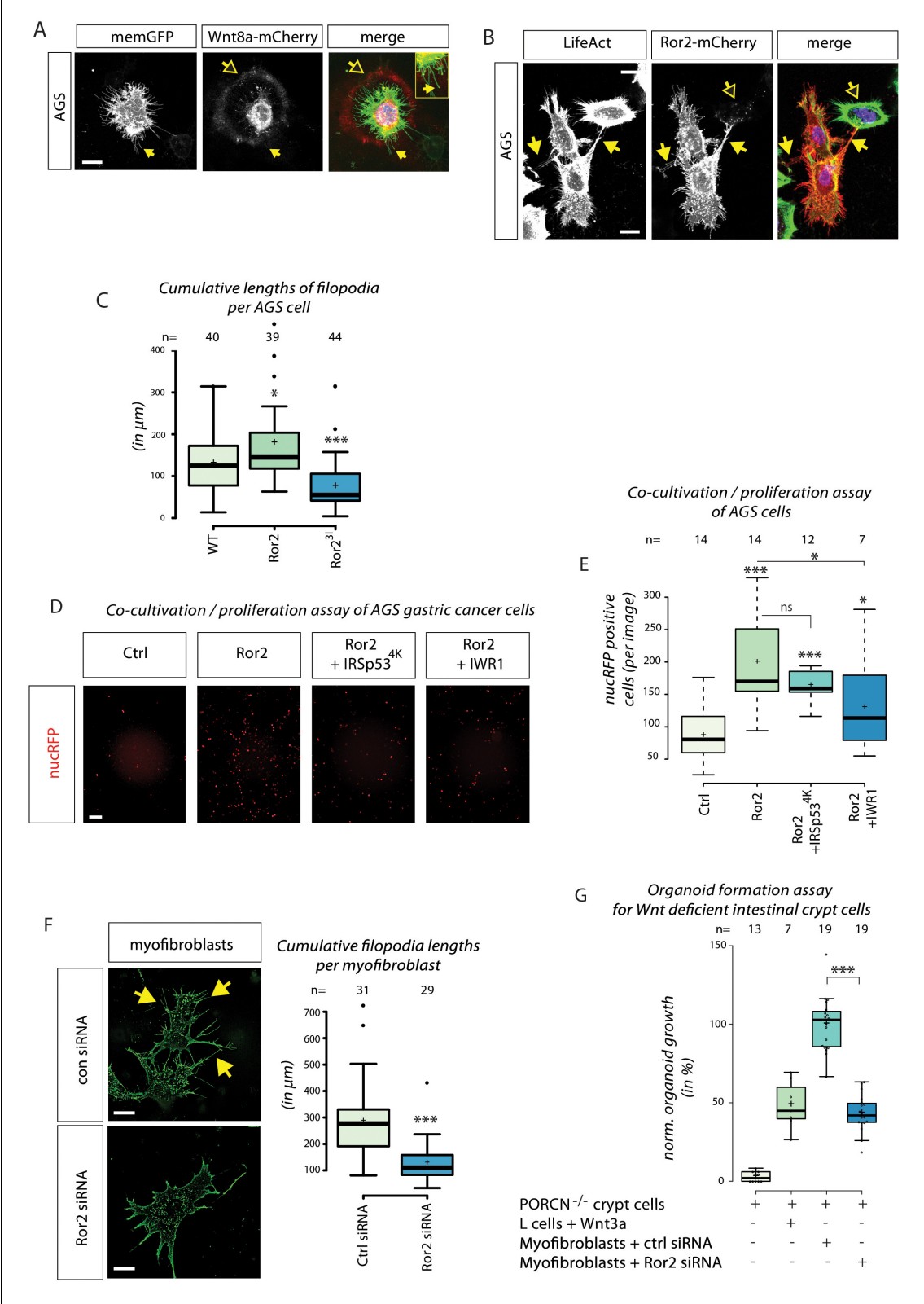

**Figure 7.** Importance of ror2-dependent cytonemes in gastric cancer cell proliferation and formation of intestinal crypt cell organoids. (**A**) Confocal Z-projection of live AGS cells transfected with memGFP and Wnt8a-mCherry. Yellow arrows mark a cytoneme and yellow outlined arrows highlight circular Wnt8a-mCherry spots around the source cell. (**B**) Confocal Z-projection of fixed AGS cells transfected with Ror2-mCherry and stained with LifeAct. Yellow arrows mark filopodia connections to adjacent cells and yellow outlined arrows highlight Ror2-mCherry clusters in a non-transfected

*Figure 7 continued on next page*

*Figure 7 continued*

adjacent cell. (**C**) Boxplot of cumulative filopodia lengths for AGS cells transfected with an empty plasmid, Ror2 or Ror2[3l]. (**D, E**) Proliferation assay of nucRFP transfected AGS cells after a 48 hr co-cultivation with cells transfected with indicated constructs. (**D**) Fluorescent images that were subjected for cell counting. (**E**) Average nucRFP cells per image. (**F**) Fluorescent images of purified intestinal myofibroblasts transfected with control siRNA (n=31) or Ror2 siRNA (n=29) were quantified for cumulative filopodia length per cell. (**G**) Boxplot of the number of formed organoids of PORCN-deficient intestinal crypt cells co-cultured with the indicated cell population for 3–4 days. Organoid numbers were normalized to crypt cells transfected with ctrl siRNA. *** $P < 0.001$, ** $P < 0.01$, * $P < 0.05$. One-way ANOVA confirmed significant differences between treatments with a 95% confidence interval of (**C**) $p=3.6151*10^{-7}$, F value=16.8, df=122; (**D**) $p=6.67274*10^{-5}$, F value=9.4, df=46; (**F**) $p=3.32078\ 10^{-06}$, F value=26.5, df=59; and (**G**) $p=1.42034*10^{-18}$, F value=106.9, df=41.

DOI: https://doi.org/10.7554/eLife.36953.018

The following figure supplement is available for figure 7:

**Figure supplement 1.** Gastric cancer cells and murine myofibroblasts use cytonemes to transport Wnt protein.

DOI: https://doi.org/10.7554/eLife.36953.019

*2018*). The intestinal myofibroblasts form a large number of filopodia (*Figure 7F*). The formation of filopodia is inhibited by siRNA-mediated knock-down of Ror2. We used an organoid formation assay to analyze the requirement of Wnt-signaling filopodia in the intestinal crypt. We used Wnt-deficient Porcn$^{-/-}$ crypt cells, co-cultivating them with Wnt3a-secreting L cells, to grow Wnt-deficient crypt organoids. Co-culture of WT myofibroblasts with Wnt-deficient crypt cells leads to induction and maintenance of crypt organoids (*Kabiri et al., 2014*). These myofibroblasts extend filopodia to engulf crypt organoids (*Figure 7—figure supplement 1C*). When Ror2 was knocked down in the Wnt-producing myofibroblasts, we observed a substantial decrease in the number of organoids (*Figure 7G*). This suggests that the transport of Wnt signals on Ror2-dependent cytonemes from the myofibroblasts is crucial for the induction and maintenance of the intestinal crypt. We conclude that cytonemes are vital for Wnt protein dissemination in vertebrates and that their appearance is regulated by Ror2-mediated PCP signaling.

## Discussion

### Cytonemes transport signaling molecules

Cytonemes are actin-based filopodia that transport an array of signaling molecules and their receptors, facilitating juxtacrine signaling. Their appearance is highly dynamic during development (*Gradilla and Guerrero, 2013*; *Kornberg and Roy, 2014*; *Stanganello and Scholpp, 2016*). A continuous adjustment of the number, length, rate of formation, direction, and retraction of cytonemes is crucial to regulate the exchange of signaling proteins within a tissue. However, how cytoneme formation is controlled remains unclear. Cytonemes change in response to signaling protein levels in the source cell (*Sato and Kornberg, 2002*; *Stanganello et al., 2015*), suggesting that cytoneme formation is linked to signal production. A signaling pathway that influences cytoneme emergence is the Wnt signaling network. We have found that cytoneme initiation is linked to Wnt8a production. Wnt8a activates the PCP signaling pathway in ligand-producing cells and, in turn, PCP induces cytonemes, which can be loaded with Wnt protein. Data from *Drosophila* show that cytonemes require the PCP components Prickle and Vangl, which are essential for Fgf- and Dpp-positive cytonemes in the air sac primordium (*Huang and Kornberg, 2016*). Prickle/Vangl modulate heparin proteoglycan content in the extracellular matrix to allow cytoneme outgrowth. By contrast, we found that Wnt8a activates the PCP pathway via interaction with the Ror2 receptor to initiate cytoneme formation. Our in vivo analysis suggests that activation of Ror2 in the zebrafish embryonic margin is necessary and sufficient for the formation of Wnt cytonemes, as other filopodia remain unperturbed. We conclude that Ror2 is a specific cytoneme regulator.

### Ror2 in filopodia formation

Ror2 is a member of the family of orphan receptor tyrosine kinases, possessing an extracellular Fzd-like CRD, a cytoplasmic tyrosine kinase domain, and a proline-rich domain (PRD) (*Yoda et al., 2003*). Ror2 plays crucial roles in developmental morphogenesis in vertebrates. Ror2-deficient mice exhibit skeletal, genital, and cardiovascular abnormalities caused by disrupted CE movements during

gastrulation (*Oishi et al., 2003*; *Takeuchi et al., 2000*). Ror2 is required for filopodia formation and XWnt5a-induced cell migration in *Xenopus* (*Nishita et al., 2006*). Irrespective of stimulation by Wnt proteins, ectopic expression of Ror2 can induce filopodia formation by actin polymerization via coupling of the Ror2-PRD to the actin-binding protein Filamin A. Our in vitro data show that Ror2 induces filopodia formation in PAC2 fibroblasts, myofibroblasts and gastric cancer cells. Moreover, we show that Ror2 is crucial for Wnt cytoneme formation in vivo, and it is known to contribute to cytoskeleton remodeling and JNK activation in migrating cells (*Oishi et al., 2003*). We have previously shown that Wnt cytonemes require Cdc42 function to stabilize the intracytonemal actin skeleton (*Stanganello et al., 2015*). Here, we demonstrate that Ror2-dependent cytonemes also require Cdc42 function for the formation and maintenance of the actin scaffold. Our data are supported by an analysis of CE movement in *Xenopus*: co-expression of Cdc42$^{T17N}$ rescues XRor2-induced CE alterations (*Hikasa et al., 2002*), suggesting that Ror2 regulates Cdc42-dependent processes.

## Ror2 and Wnt ligands

Ror2 acts as a receptor for Wnt5A in mice (*Mikels et al., 2006*). In *Xenopus*, Wnt5A/Ror2 activates the PCP signaling pathway, including transcription of ATF2 and PAPC (*Schambony and Wedlich, 2007*). Although there is in vitro evidence that multiple Wnts can associate with the CRD domain of Ror2, only a few Wnts have been demonstrated to elicit Ror2 activation in vivo. For example, Wnt1 and Wnt3a bind to the CRD domain of Ror2 (*Billiard et al., 2005*) but neither altered receptor activity as assessed by levels of Ror2 autophosphorylation. *Xenopus* Wnt8 binds to the ectodomain of Ror2 (*Hikasa et al., 2002*) but it is not known whether XWnt8 induces Ror2 signaling. In zebrafish, evidence suggests that Wnt11 is a potential binding and signaling partner for Ror2 (*Bai et al., 2014*). Wnt11 activates Ror2 to modulate CE during zebrafish gastrulation. We have observed that Wnt8a co-localizes with Ror2 in cytonemes. Although tagged constructs have been examined at very low concentration, this does not necessarily imply correct subcellular localization. However, our results are in line with studies on endogenous Wnt8a localization on cytonemes (*Stanganello et al., 2015*) and with studies on Ror2 localization on filopodia in mouse (*Paganoni and Ferreira, 2003*; *Laird et al., 2011*) and in *Xenopus* (*Nishita et al., 2006*; Brinkmann et al., 2016). We have further observed that Wnt8a binds and activates Ror2 signaling in vitro and in vivo. It is possible that other β-catenin-independent Wnts, such as Wnt5a and Wnt11, might induce Ror2 cytonemes to regulate β-catenin Wnt ligand trafficking and thus β-catenin signaling. Wnt5A and Wnt11 are known to regulate the dorso-ventral patterning of the neural tube and somites in mice (*Andre et al., 2015*). However, the observed patterning defect and AP axis shortening in Wnt5A/Wnt11 double-knockout mice was explained by migration alteration of the axial mesoderm.

## Interaction of the Wnt signaling branches

Binding of Wnt proteins to their cognate Frizzled receptors activates several distinct signaling pathways, including the canonical Lrp6/β-catenin-dependent and the non-canonical Ror2/β-catenin-independent PCP pathways (*Niehrs, 2012*). According to the conventional classification, Wnt1, Wnt3a, and Wnt8a belong to the β-catenin-dependent Wnt signaling proteins, whereas Wnt5a and Wnt11 are representatives of the β-catenin-independent Wnt signaling proteins (*Kikuchi et al., 2011*). Activation of the Ror2/PCP signaling branch is assumed to repress the β-catenin signaling branch (*Niehrs, 2012*). Both Wnt signaling branches use the same hub proteins as intracellular effectors, which may lead to competition and mutual repression. For example, Wnt5a, which preferentially activates PCP signaling, competes with Wnt3a for Fzd2 and thus inhibits the β-catenin-dependent pathway (*Sato et al., 2010*). Furthermore, in *Caenorhabditis elegans*, Ror2 is thought to act by sequestering β-catenin Wnt ligands, a function that is independent of its intracellular domain (*Green et al., 2007*). In tissue culture, intracellular Ror2 signaling represses β-catenin signaling via its tyrosine kinase activity (*Mikels et al., 2006*) and its interaction with Dvl (*Witte et al., 2010*). Furthermore, intracellular binding partners such as Tak1 interact with the C-terminal region of Ror2 to further inhibit Wnt/β-Catenin signaling (*Winkel et al., 2008*). However, recent work reported that Ror2 signaling is able to enhance β-catenin-mediated signaling and proliferation in several tumor types (*Rasmussen et al., 2013*; *Roarty et al., 2017*; *Yan and Lin, 2008*). These apparently contradictory results could be explained by cross-regulation of Ror2 and the β-catenin signaling being influenced by tissue heterogeneity. This might occur, for example, during tumor progression (*Roarty et al.,*

*2017*) when depending on the repertoire of Wnt signaling components present, Ror2-expressing tumor cells regulate the spatial distribution, duration, and amplitude of Wnt/β-catenin signaling within a tumor. Our data provide a mechanistic explanation for this effect. Although we cannot rule out the possibility of Ror2 activation within the Wnt source cells, we hypothesize that the Ror2/PCP signaling pathway is activated in an autocrine fashion on the basis of our observations of the behavior of individual cells in cell culture and in the embryo. We further provide evidence that Ror2/PCP-dependent signaling is crucial for cytoneme emergence in Wnt source cells. Then, cytonemes transmit Wnt ligands to neighboring cells to activate juxtacrine β-catenin signaling. We demonstrate the importance of separating Wnt-producing from Wnt-receiving cells by showing that Ror2-mediated Wnt transport is required for controlling cell proliferation in AGS gastric cells. AGS cells express Wnt ligands such as Wnt1, display a high endogenous β-catenin level, and show enhanced proliferation (*Mao et al., 2014*).

The anti-cancer drug salinomycin inhibits β-catenin signaling by inducing degradation of the Wnt co-receptor Lrp6, resulting in reduced proliferation. Autonomous activation of PCP signaling via Ror2 represses β-catenin signaling and reduces proliferation in AGS cells (*Yan et al., 2016*), suggesting that activation of PCP signaling could be used as a further strategy to inhibit uncontrolled proliferation of gastric tumors. However, gastric tumors display a high degree of tissue heterogeneity and we find that Ror2 can also enhance β-catenin signaling and consequently cell proliferation. Our experimental approach is fundamentally different to that of the former analysis: Wnt-producing cells and Wnt-receiving cells are treated separately. We find that Ror2 activation in the Wnt-producing cells increases proliferation in the Wnt-receiving cells. This hypothesis is supported by recent findings showing that Wnt-Ror2-positive mesenchymal cells promote gastric cancer cell proliferation if co-cultivated (*Takiguchi et al., 2016*).

## Cytonemes are a general carrier for Wnt protein dissemination

Here, we describe Ror2/PCP-induced cytonemes as transport carrier for Wnt8a in zebrafish. In cell culture experiments, we use PAC2 fibroblasts and HEK293T cells to provide further evidence for the importance of Ror2-dependent cytonemes in Wnt trafficking. In addition, we show that human gastric cancer cells AGS, which primarily express the Wnt ligand Wnt1, process paracrine Wnt signaling via cytonemes, which are influenced by Ror2 signaling. Finally, we use murine intestinal stroma cells, which express Wnt2b to maintain the Wnt gradient operating in the intestinal crypt (*Aoki et al., 2016*; *Kabiri et al., 2014*). We provide further evidence that Wnts from intestinal stroma utilize Ror2-dependent cytonemes for their transport. In summary, we show that autocrine PCP pathway activation via Ror2 induces Wnt cytonemes in the Wnt source cell to transmit Wnt to the neighboring cell to activate paracrine Wnt/β-catenin signaling. We propose cytonemes as an general mechanism for the mobilization of Wnt ligands in tissue homeostasis as well as in development in vertebrates.

## Materials and methods

### Plasmids

The following plasmids were used: pCS2 + zfWnt8aORF1 (Addgene 17048), pCS2 + zfWnt8aORF1-GFP, pCS2 + zfWnt8aORF1-mCherry (*Stanganello et al., 2015*), pCS2 + xWnt5 a-GFP, pCS2 + zfWnt5 b (*Wallkamm et al., 2014*), xRor2-mCherry (*Feike et al., 2010*), xRor2[3I] (*Casella et al., 1981*), mRor2-dCRD-GFP, pCS2 +Fz7 a-CFP, pcDNA3-EGFP-Cdc42T17N (Addgene 12976), 7xTRE Super TOPFlash-NLS-mCherry (*Moro et al., 2012*), pmKate2-f-mem (Evrogen), GPI-anchored mCherry in pCS2+ (*Scholpp et al., 2009*), IRSp53[4K] (*Casella et al., 1981*). To generate the pCS2 + xRor2 construct, the open reading frame of xRor2-mCherry was inserted into the ClaI/XhoI site of pCS2+.

### Maintenance of fish

Breeding zebrafish (*Danio rerio*) were maintained at 28°C on a 14 hr light/10 hr dark cycle (*Brand et al., 2002*). To prevent pigment formation, embryos were raised in 0.2 mM 1-phenyl-2-thio-urea (PTU, Sigma, St Louis, MO 63103, USA) after 24 hpf. The data we present in this study were acquired with wild-type zebrafish (AB2O2) as well as with the transgenic zebrafish line tg(−6gsc:EGFP-CAAX) (*Smutny et al., 2017*). All animal work (zebrafish husbandry and experimental

procedures) were undertaken under project and personnel licences granted by the UK Home Office under the United Kingdom Animals (Scientific Procedures) Act, in accordance with The University of Exeter's ethical policies and were approved by the University of Exeter's Animal Welfare and Ethical Review Body, and in accordance with the German law on Animal Protection approved by the Local Animal-Protection Committee (Regierungspräsidium Karlsruhe, Az.35–9185.64) and the Karlsruhe Institute of Technology (KIT).

## Cell culture experiments

Experiments were performed in zebrafish PAC2 fibroblasts derived from 24-hr-old zebrafish embryos cultivated at 28°C without additional $CO_2$ supply, primary gastric adenocarcinoma cells (AGS), gastric tubular adenocarcinoma liver metastasis cells (MKN28 and MKN7 [*Motoyama et al., 1986*]) and human embryonic kidney cells (HEK293T; CRL-1573) cultivated at 37°C with 5% additional $CO_2$ supply. PAC2 were maintained in Leibowitz-15 media, HEK in DMEM, AGS and MKN28 in RPMI 1640, all supplemented with 10% fetal bovine serum, 1% L-glutamine (2 mM) and 1% penicillin/streptomycin. All of the materials used for cell culture were purchased from Life Technologies, Gibco.

For transfection experiments, FuGENE HD Transfection Reagent (Promega) was used and cells were imaged after 48 hr. For co-culture experiments, transfected cells were incubated for 24 hr, detached by Trypsin-EDTA (0.05%) and incubated in a mixed population for another 48 hr before image acquisition.

Assays for SuperTOPFlash (STF) TCF/Wnt reporter expression and proliferation required initial co-cultivation of two distinct cell populations. Cells were transfected with pDest7xTCF-NLS-mCherry or pCS2 +nucRFP plasmids, respectively, and incubated for 24 hr, detached by trypsin-EDTA (0.05%) and further incubated in a mixed population for another 72 hr before image acquisition. For one replication, seven 10x magnification images were taken per sample with identical laser settings. Image locations were saved by the Mark and Find microscope feature to reproduce similar scanning setups. For measuring TCF/Wnt reporter activation, fluorescent nuclei were processed using the Dot-Plugin in Imaris and the average grey value of the nuclei was determined or fluorescent nuclei were counted to measure cell proliferation.

For the chemical treatment, cell cultures were treated with GTPase inhibitor ML141 10 mM (Merck Millipore) or 50 μM IWR-1 (Sigma) to antagonise the Wnt signaling pathway. For staining, cells were fixed with 4% PFA at RT, washed with PBS. Cells were incubated with 50 μg/ml phalloidin (P1951, Sigma) and 10 μg/ml DAPI (D9542, Sigma).

## Organoid formation of intestinal crypt cells

Myofibroblasts were prepared from C57BL/6-*Tg(Pdgfra-cre)1Clc/J/Rosa*$^{mTmG}$ mice and cultured as previously described (*Greicius et al., 2018*). As confluence of cultured cells was reaching 80%, they were transfected with respective the siRNA (Dharmacon mouse ROR2 siRNA pool Cat#LQ-041074-00-0002, four siRNAs combined in equal parts at 10 nM) using siRNAmax reagent (Invitrogen Cat#13778–030). On day 2 post-transfection, myofibroblasts were mixed with *Porcn*-deficient intestinal epithelial cells and cultured using RSPO1-supplemented medium. Organoid counting was performed at the time point when the group containing no stromal cells had no surviving organoids left (the end of day 3/beginning of day 4 of co-culture). siRNA-transfected cells were imaged using the OlympusLive Imaging system IX83. Acquired 3D image stacks were de-convoluted using cellSens Dimension (Olympus) and are presented as maximum intensity projections.

## Automated filopodia analysis software

Cells and their attached filopodia were automatically detected in the RFP channel (mem-mCherry) of the acquired images (*Figure 4—figure supplement 1A*). The images were initially filtered using a Gaussian low-pass filter ($\sigma^2 = 1$) and subsequently used detect the cell body as well as the cell's filopodia (*Figure 4—figure supplement 1B*). For the filopodia detection, we used an objectness filter ($\sigma = 1, \ \alpha = 1, \beta = 1, \gamma = 0.003, \ N = 2$) that emphasized line-like structures based on the eigenvalues of the Hessian matrix at each pixel location (*Figure 4—figure supplement 1C*, [*Antiga, 2007*]). The obtained edge-enhanced image was then binarized (*Figure 4—figure supplement 1D*) using a local adaptive threshold filter that set pixels to foreground if their intensity value was larger than a regional mean intensity minus a multiple of the regional intensity standard deviation and otherwise

to background (window radius = 200, std. dev. multiplier = 1). To segment the cell body, we applied the local adaptive threshold (window radius = 200, std. dev. multiplier = 1) on the smoothed input image (*Figure 4—figure supplement 1E*) and subsequently used a morphological opening operation (kernel radius = 2) to get rid of noise and remaining filopodia parts (*Figure 4—figure supplement 1F*, [*Soille et al., 2011*]). The cell body was given by the largest connected component in the opened binary image. The segmentation mask of the cell body including filopodia was then constructed by combining the binarized edge-enhanced image with the binary cell body image (*Figure 4—figure supplement 1G*). The combined cell image was subsequently skeletonized to identify potential filopodia tips at the end points of the skeleton (*Figure 4—figure supplement 1H*). All of the above-mentioned preprocessing steps were implemented in the open-source software tool XPIWIT (*Bartschat et al., 2016*).

The preprocessing results were then imported into a dedicated MATLAB tool that was developed to validate, correct and analyze this kind of images (*Figure 4—figure supplement 1I,J*). In order to trace filopodia from the identified tips to the cell body automatically, we used an adapted livewire algorithm (*Barrett and Mortensen, 1997*). The output of the objectness filter was used as an edge map (*Figure 4—figure supplement 1C*) on which the livewire algorithm tried to find a maximally scoring path from the tip of the filopodium to the center of the cell body. Based on the segmentation of the cell body (*Figure 4—figure supplement 1F*), the automatic tracing was stopped as soon as the cell body was reached. The interactive user interface was then used to add, remove and correct both segmentation masks and detected filopodia on a per-cell basis. For each cell's filopodia, we calculated the Euclidean distance along the the path from the tip to the cell body. The same image preprocessing and tracing was applied to semi-automatically extract filopodia in 3D confocal images (*Figure 4—figure supplement 1K*). For 3D images, however, the start and end points of the filopodia were provided by the user via a graphical user interface and the livewire approach was applied twice: first on an axial maximum intensity projection (z) to obtain the lateral path (xy), then the axial positioning of the filopodium was obtained by searching for the highest scoring path between the provided start and end points solely in the z-direction (*Figure 4—figure supplement 1K*). Multiple automatically traced filopodia can then be exported and used to obtain average statistics for all filopodia of interest.

## lsFCS

For the FCS measurements, a home-built confocal microscope was used as previously described (*Dörlich et al., 2015*), with slight modifications. We used a water immersion objective (HCX PL APO CS 63x/1.2, Leica, Wetzlar, Germany) instead of an oil immersion objective; the multimode fiber, which acts as a confocal pinhole, was modified accordingly to ensure a pinhole size of 1 AU. Data were collected for 390 s by continuously scanning the focus perpendicularly through the membrane. Each scan line consisted of 100 pixels, with a step size of 100 nm. eGFP was excited with a 488 nm continuous wave (cw) laser and mCherry with a 561 nm cw laser. After splitting the fluorescence signal into two color channels by using a 555 nm dichroic filter, 525/50 (eGFP) and 600/37 (mCherry) band pass filters were used for detection. To avoid artefacts in the correlation curves caused by scanner flyback and wavelength switching, the membrane was always kept in the center of the field of view. The intensity data were arranged as an *x-t* pseudo image, and the intensities of those pixels containing membrane fluorescence were integrated to obtain an intensity time trace for correlation analysis, as described earlier (*Hunter, 2007*).

## Functional analysis

The injection of mRNAs and Morpholino oligomers was performed according to the description in the text and in *Mattes et al. (2012)*. Ror2 MO was used in a 0.5 mM concentration (5'-CAGTGTAA-CAACTTCCAAACTCTCC −3') (Gene Tools, Philomath, OR 97370, USA). Capped and in vitro transcribed mRNA (mMessage Machine Kit, Ambion) was microinjected into one cell or into the yolk for ubiquitous expression or into one of 16 blastomeres to generate cell clones. Embryos were incubated at 28°C until used for image acquisition or fixed for whole-mount mRNA in situ hybridization (ISH).

For ISH, *hgg*, *ntl*, *axin2*, *fgf8a* and *pax6a* digoxigenin- and fluorescein-labeled probes were prepared from linearized templates using an RNA labelling and detection kit (Roche) as described by

*Scholpp and Brand (2003)*. Images were taken on an Olympus SZX16 microscope equipped with a DP71 digital camera by using Cell D imaging software.

For real-time quantitative PCR (RT-qPCR), 50 embryos each were lysed in 1 ml TriZol (Sigma), and total RNA was prepared using the Direct-zol RNA Mini Prep Kit from Zymo Research. cDNA was prepared using MMLV reverse transcriptase from Promega and analysed in a Real-Time PCR system from LifeTechnologies (ABI StepOnePlus). Primers with the following sequence were used: *beta-actin* (5′-CCTTCCTTCCTGGGTATGG-3′; 5′-GGTCCTTACGGATGTCCAC-3′), *axin2* (5′-CAATGGAC-GAAAGGAAAGATCC-3′; 5′-AGAAGTACGTGACTACCGTC-3′), *lef1* (5′-CAGACATTCCCAATTTCTA TCC-3′; 5′-TGTGATGTGAGAACCA ACC-3′). Results were analysed using the ΔΔCT method.

## Simulations

Computer simulations have been implemented in Python using the libraries numpy, scipy and matplotlib (*Hunter, 2007*) and are based on a simulation of cytoneme-based ligand distribution described in detail in *Stanganello et al. (2015)*. The modelling of the tissue is split up into two sub-problems (1) the correct modelling of the dynamically forming tissue and (2) the morphogen transport through that tissue. The tissue dynamics are modelled by a two-dimensional non-periodic plane of discrete cells. The simulated area is 1,000 μm by 1,000 μm in size and each cell occupies a circle of 8 μm radius. Possible cell positions are precomputed and kept fixed during the course of the simulation. Cells can change positions by moving along those precomputed positions. In every simulation step ($\Delta t = 1$ s), it is possible for each individual cell to perform actions with predetermined probabilities, which are directly derived from experimental measurements.

Possible actions are: (i) signaling from the producing tissue, (ii) cell insertion by division or intercalation of the receiving tissue, (iii) cell migration — directed or non-directed nearest neighbor swapping of the receiving tissue, and (iv) morphogen decay in the receiving tissue. Two different cell types are distinguished. The marginal, Wnt active cells produce morphogen and are able to deposit morphogen via cytonemes into cells of the receiving tissue. Once a signaling event is accepted for a marginal cell, an angle is randomly chosen from the respective distribution determined in *Stanganello et al. (2015)*. The length is randomly chosen from a Gaussian distribution with the peak value $l_{filo}$. A virtual filopodium is formed with those values originating from the marginal cell. If the tip of the filopodium ends up in the vicinity of the surface of a receiving cell ($\pm 2$ μm) the Wnt content of that cell is increased, otherwise the filopodium is deleted.

### Probabilities used in the simulation

| | Signaling probability ($p_{filo}$) |
|---|---|
| WT | $\frac{1}{30}$ |
| Ror2 | $1.61 \cdot \frac{1}{30} = 0.0537$ |
| Ror2[3I] | $0.338 \cdot \frac{1}{30} = 0.0113$ |

Tissue expansion during neural plate patterning is driven by the intercalation of cell layers during the thinning out of the cell sheet as well as during cell division. As we are only modelling one cell layer, both of these processes can be incorporated into a single action, namely cell insertions. If a cell insertion event is accepted in a receiving cell, a path to the nearest empty grid spot is obtained and all cells are subsequently moved along that path. The emerging empty grid spot is then filled with a randomly chosen cell with the same distance $\pm 6$ μm from the marginal cells. To include the highly dynamic intermingling of the cells during that process, an additional action — cell migration — is added. This action allows for a cell to swap places with a randomly chosen nearest neighbor cell. Morphogen decay inside the receiving tissue is implemented by an action that decreases the Wnt content of the cell once accepted.

## Luciferase reporter assay in *Xenopus* embryos

For the ATF2 luciferase reporter assay, four-cellstage *Xenopus* embryos were injected into animal ventral blastomeres along with the 100 pg ATF2-luciferase reporter plasmid and 10 pg TK-Renilla-

luciferase reporter plasmid. The reporter plasmids were injected alone or together with 500 pg of the respective synthetic mRNAs. Luciferase reporter assays were carried out using triplicates of five-gastrula-stage (st.12) embryos lysed to measure luciferase activity using the dual luciferase system (Promega).

## Image acquisition

For confocal analysis, live embryos were embedded in 0.7% low melting agarose (Sigma-Aldrich) dissolved in 1x Ringer's solution. Images of cells and embryos were obtained with a Leica TCS SP5 X or SP8X confocal laser-scanning microscope using 20x or 63x dip-in objective or, for the kinase library screen, a Leica DMI600SD with 20x objective. Image processing was performed with Imaris 9.1 software (Bitplane AG, Switzerland). Filopodia and cytoneme measurements from confocal z-stacks of living embryos was performed via the semi-automated filopodia analysis software described before. Cell culture quantifications were carried out using Fiji software. Roundness of notochordal embryo cells was determined by calculating the width to length ratio of each cell in Fiji.

## Statistical analysis

All experiments were carried out at least in biological triplicates if not indicated otherwise. Significance was calculated by Student's t-test and asterisks are used to indicate p values. One-way ANOVA was used to analyse groups of experimental data. All groups are random samples from the same population. Variances are similar across treatments and residuals are normally distributed. P values, F values and degrees of freedom (df) are indicated. Box plots: centre lines show the medians; box limits indicate the 25th and 75th percentiles as determined by R software; whiskers extend 1.5 times the interquartile range from the 25th and 75th percentiles, outliers are represented by dots.

## Acknowledgements

This project was funded by the Living Systems Institute, the University of Exeter and the Boehringer Ingelheim Foundation (awards to SS). Studies in the DMV lab are supported by the National Research Foundation of Singapore and National Medical Research Council under its STAR Award Program. JR and AS were supported by the Impuls- und Vernetzungsfond of the Helmholtz Association. GUN was funded by the Deutsche Forschungsgemeinschaft (SFB 1324, projects A6 and Z2, GRK2039) and the Helmholtz Association Program STN. For technical assistance, we would like to thank Donya Shapoori (RTqPCR analysis), Julia Schuller and Melanie Merkel (Fzd7/Ror2 interaction studies). We also thank Trevor Dale and Toby Phesse (ECSCRI, Cardiff University) for providing the gastric cancer cell lines; Francesco Argenton (University of Padova), Steve Wilson (UCL), Masa Tada (UCL), Jochen Wittbrodt (University of Heidelberg) and Yashuiro Minami (Kobe University) for providing plasmids and Gáspár Jékely (LSI Exeter) for comments on the manuscript. We would like to thank the Aquatic Resources Centre (ARC) and the BioImaging Centre, Exeter for excellent technical support.

## Additional information

### Funding

| Funder | Grant reference number | Author |
| --- | --- | --- |
| Helmholtz Association | Impuls- und Vernetzungsfond | Jakob Rosenbauer Alexander Schug |
| Deutsche Forschungsgemeinschaft | GRK2039 | Gerd Ulrich Nienhaus |
| University of Exeter | Living Systems Institute: start-up | Steffen Scholpp |
| Boehringer Ingelheim Fonds | Exploration | Steffen Scholpp |
| Deutsche Forschungsgemeinschaft | Scho847-5 | Steffen Scholpp |

| Helmholtz Association | STN program | Gerd Ulrich Nienhaus |

The funders had no role in study design, data collection and interpretation, or the decision to submit the work for publication.

## Author contributions

Benjamin Mattes, Conceptualization, Data curation, Software, Formal analysis, Validation, Investigation, Visualization, Methodology, Writing—original draft; Yonglong Dang, Data curation, Software, Formal analysis; Gediminas Greicius, Data curation, Formal analysis, Visualization; Lilian Tamara Kaufmann, Benedikt Prunsche, Jakob Rosenbauer, Suat Özbek, Data curation; Johannes Stegmaier, Ralf Mikut, Software; Gerd Ulrich Nienhaus, Alexander Schug, David M Virshup, Data curation, Supervision; Steffen Scholpp, Conceptualization, Data curation, Formal analysis, Supervision, Funding acquisition, Investigation, Visualization, Writing—original draft, Project administration, Writing—review and editing

## Author ORCIDs

Benjamin Mattes (iD) https://orcid.org/0000-0001-5286-9347
Johannes Stegmaier (iD) http://orcid.org/0000-0003-4072-3759
Gerd Ulrich Nienhaus (iD) http://orcid.org/0000-0002-5027-3192
David M Virshup (iD) http://orcid.org/0000-0001-6976-850X
Steffen Scholpp (iD) http://orcid.org/0000-0002-4903-9657

## Decision letter and Author response

Decision letter https://doi.org/10.7554/eLife.36953.024
Author response https://doi.org/10.7554/eLife.36953.025

## Additional files

### Supplementary files

• Transparent reporting form
DOI: https://doi.org/10.7554/eLife.36953.020

### Data availability

All of the data supporting this paper is available via the Dryad repository (https://dx.doi.org/10.5061/dryad.38q5pc1)

The following dataset was generated:

| Author(s) | Year | Dataset title | Dataset URL | Database, license, and accessibility information |
| --- | --- | --- | --- | --- |
| Schlopp S | 2018 | Data from: Wnt/PCP controls spreading of Wnt/$\beta$-catenin signals by cytonemes in vertebrates | https://dx.doi.org/10.5061/dryad.38q5pc1 | Available at Dryad Digital Repository under a CC0 Public Domain Dedication |

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
