## [Decision Letter]

Thank you for submitting your article "Cytonemes bridge Wnt autocrine and paracrine signaling systems" for consideration by *eLife*. Your article has been reviewed by three peer reviewers, and the evaluation has been overseen by a Reviewing Editor and Didier Stainier as the Senior Editor. The following individual involved in review of your submission has agreed to reveal his identity: Ajay B. Chitnis (Reviewer #1).

The reviewers have discussed the reviews with one another and the Reviewing Editor has drafted this decision to help you prepare a revised submission.

Summary:

The reviews are very positive and agree that the findings are interesting and should make an important contribution to the field. There are some issues for the authors to consider in a revised version. Most are related to presentation and statistical analysis; the major issue is that because the imaging experiments are based on over expression, it is essential that the authors demonstrate that the measurements, observations and interpretations are not skewed by the over expression. One way would be to analyze fusion proteins expressed at physiological levels using endogenous regulatory sequences. If this cannot be accomplished in a several month time frame, perhaps it would be possible to assess localization of protein distribution at low expression levels under conditions of limited RNA abundance.

Specific requests (many of which do not require additional experiments):

1) The evidence for clustering of Wnt8a and Ror2 could be stronger. The authors do recognize the risk of non-specific clustering due to overexpression, and so test a CRD-deleted version of Ror2, which seems to result in less clustering. Nevertheless, the example plots in Figure 2B show only partial regions of single cells, and so it is difficult to assess the correlation or lack thereof across construct types. A more compelling demonstration would show these same sorts of data for multiple cells with appropriate statistical analyses to assess changes in correlation strength, and reporting for sample sizes. The lsFCS analysis would seem to also benefit from a comparison of CRD-deleted Ror2 to native Ror2.

2) The presentation of Wnt8a transport and Lrp6 clustering in Figure 5 and Figure 5—figure supplement 1 is confusing. Delineation of the target cell from brightfield comparison does not seem to match evident cell boundaries –.…is this a consequence of a different focal plane? The statement that of clusters "dissolve" in the membrane of the extending cell is unclear ("enter," "transit"?). Additionally the authors describe the formation of Lrp6-GFP clusters but it is not clear if clustering dynamics have been observed or only fully formed clusters. Finally, to better demonstrate that Wnt8 is actually endocytosed it would seem useful to display orthogonal views to illustrate the presence of reporter within the target cell.

3) Statistical analyses need to be improved. Besides the issue raised in #1 above, the authors should throughout provide test statistics with degrees of freedom (and thus indication of sample size) in addition to P values. Presentations of significant differences for means should also include overall ANOVA with corrections for multiple comparisons. Since experimental treatments seem to be sometimes compared to controls and sometimes to one another, it would be preferable to simply indicate means that are not significantly different from one another in post hoc analyses (e.g., using letter codes). It is also unclear whether data obtained meet the assumptions of the statistics used. For example, in the second paragraph of the subsection “Ror2 dependent cytonemes in cancer cells and intestinal organoids”, it states that IRSp53-4K/Ror2 cells are significantly reduced in proliferation compared to Ror2 cells. Yet a major difference illustrated in the plot would seem to be a change in variance, which would likely require a transformation for analyzing appropriately, and it is not clear from the plot what the significance level of this comparison might be. Similarly in the first paragraph of the subsection “Ror2 induces Wnt-cytonemes during zebrafish neural patterning”, it notes that significant differences in filopodia numbers and lengths were observed across stages, yet Figure 4B does not clearly illustrate stage differences in lengths, only numbers. Finally, it is not clear why the authors present individual data points in 7G but only histograms or box plots elsewhere.

4) The ways in which authors distinguish cytonemes from other filopodia are not clear. The data presented in Figure 4F to argue that are Ror2 regulates a specific class of Wnt8a+ cytonemes also need clarification. For instance, are the number of Wnt8+ and Wnt8- cytonemes (filopodia?) being compared within cells, or totals across cells? Given the alternative "states" of cytonemes, comparisons involving chi-square analyses would seem more appropriate. Likewise the example cells illustrated in Figure 4E seem to be especially extreme cases, and so perhaps not representative of the overall dataset.

5) The simulation results in Figure 4G are potentially interesting as a follow up to the prior work. However, the authors need to more clearly indicate how the changes in numbers of cytonemes used in the simulations correspond to those obtained empirically: are they over the same order of magnitude? There also should be more explanation as to whether the simulations account for variation in lengths of individual cytonemes, and changes in the frequencies of cytonemes across stages as observed in Figure 4B. Finally, the simulations would be more powerful were it possible to assign specific times (e.g., hpf) or distances (μM), rather than relative values only.

6) The authors have several useful reagents for manipulating cytonemes. But there is always a concern about off-target effects of dominant negative constructs, and the potential that other pathways could be involved. With regard to Ror2 in particular, it would be useful to know whether the Ror2 morpholino used for analyses of PCP signaling, causes a loss of cytonemes similar to that of the dominant negative. Likewise, are cell behaviors unrelated to cytoneme extension and signaling unchanged in cells manipulated for Ror2 function, IRSp53, etc.? Indeed the evidence that cytoneme formation is strictly dependent n PCP signaling is somewhat circumstantial in the sense that defects in convergent extension are consistent with PCP defects, and Ror2 is known to function in PCP signaling; the authors might therefore be more circumspect about potential roles for other pathways or other ways in which Ror2 might function.

7) The authors describe a computational model that supports their experimental observation. While this adds to strength of the paper, I felt not enough detail about this model was provided for me to evaluate it.

8) Figure 2 presents data to demonstrate the specificity of interaction between Wnt8a-GFP and its receptor, Ror2-mcherry. With the data in Figure 2A/B, they attempt to show that these proteins interact, based on the colocalization of these proteins at the cell surface. Although they show an example of a correlative intensity tracing (Figure 2B), it is unfortunately not evident for this to be the case in the raw data shown in Figure 2A/middle panel: the fluorescence intensity of Ror2-mCherry appears uniform along the plasma membrane particularly at the sites showing the Wnt8-GFP spots. I think it is important to show not only more examples, but also to present a serious statistical analysis of how often and to what extent they detect colocalization.

Figure 2C/E shows an example of IsFCS analysis. This example is consistent with molecular association in regions of colocalization based on similar cross correlation and auto-correlation profiles and longer lag times, hence slower diffusion. As indicated above, I think it is important to show more examples, and clearly describe the number of events that where analyzed and clarify whether it is always the case that colocalization equates to complex formation.

Not clear from this analysis, is the possible influence that expression level might have on the outcomes and determinations. I think this is important particularly since it is likely that the tagged proteins are expressed at a significantly higher level than endogenous. The authors should also indicate the sensitivity of their microscopes, so that one can estimate the number of proteins that engaged as complexes.

[Editors' note: further revisions were requested prior to acceptance, as described below.]

Thank you for resubmitting your work entitled "Cytonemes bridge Wnt autocrine and paracrine signaling systems" for further consideration at *eLife*. Your revised article has been favorably evaluated by Didier Stainier (Senior Editor), a Reviewing Editor, and one reviewer.

Thank you for attending to issues that were previously raised, but we are not convinced that the revised manuscript addresses all of the substantive concerns. I would think that the following issues could be addressed without undue further effort or delay.

1) The robustness of colocalization analyses remain unclear given the apparently limited sampling. Although Pearson coefficients, as well as a different look up table, are helpful, they do not really address the issue of sampling adequacy. With conclusions based apparently on one cell per treatment it is difficult to know how representative these observations might be. Is it possible to expand the analysis to multiple/more cells?

2) Although sample sizes and overall P values have been added in places, the authors still do not report test statistics (e.g., F values) or their associated degrees of freedom, which are also important for evaluating the robustness of their inferences. Similarly they have not addressed whether their data meet the assumptions of ANOVA or t-tests, specifically that variances are similar across treatments and residuals are normally distributed.

3) The issue of whether or not fluorescent proteins are properly localized remains somewhat murky. The authors have provided additional data to show that concentrations used did not result in overt morphological defects. Yet apparently normal morphology does not necessarily imply correct subcellular localization. The analyses would have inspired more confidence if localization had been examined at very low concentrations of injected construct, or across a range of concentrations. Minimally, clear statements of caveats associated with these analyses would seem important.

---

## [Author Response]

Summary:The reviews are very positive and agree that the findings are interesting and should make an important contribution to the field. There are some issues for the authors to consider in a revised version. Most are related to presentation and statistical analysis; the major issue is that because the imaging experiments are based on over expression, it is essential that the authors demonstrate that the measurements, observations and interpretations are not skewed by the over expression. One way would be to analyze fusion proteins expressed at physiological levels using endogenous regulatory sequences. If this cannot be accomplished in a several month time frame, perhaps it would be possible to assess localization of protein distribution at low expression levels under conditions of limited RNA abundance.

To tackle this concern, we divided the experiments in two groups: to investigate the function of Ror2, we used a strategy to overexpress the construct and backed up our analysis with a dominant negative construct, Ror^3I^ (Figures 3, 4, 6, 7). Furthermore, we added a Ror2-specific knock-down approach, which confirmed our results (Figure 3, 4).

To investigate the localisation of Ror2 during cytoneme initiation we used a tagged version of Ror2, Ror2-mCherry. To rule out over expression artefacts caused by the tagged constructs, we expressed Ror2-mCherry at a sub-functional level. To better illustrate this approach, we added a titration experiment for the various concentrations investigating the phenotypic alterations after overexpression (Figure 3—figure supplement 1A). Based on this analysis, we chose an mRNA concentration for Ror2-mCherry of 25ng/µl for all following experiments. We added the amendments to the text (subsection “Cluster formation of Wnt8a and Ror2 is dependent on the CRD domain”) and the figure legends.

Specific requests (many of which do not require additional experiments):1) The evidence for clustering of Wnt8a and Ror2 could be stronger. The authors do recognize the risk of non-specific clustering due to overexpression, and so test a CRD-deleted version of Ror2, which seems to result in less clustering. Nevertheless, the example plots in Figure 2B show only partial regions of single cells, and so it is difficult to assess the correlation or lack thereof across construct types. A more compelling demonstration would show these same sorts of data for multiple cells with appropriate statistical analyses to assess changes in correlation strength, and reporting for sample sizes.

We improved the image-quality of the Ror2/Wnt8a in Figure 2A. To illustrate clustering potential of the investigated constructs, we added a high-magnification image of the co-localisation channel in fire-LUT in Figure 2B.

In addition, we calculated the corresponding Pearson Correlation Coefficients (Figure 2—figure supplement 1G). For this analysis, we used X stacks with a volume of Y µm3. This analysis demonstrate that the CRD domain of Ror2 is important for formation of the Wnt8a/Ror2 clusters.

The lsFCS analysis would seem to also benefit from a comparison of CRD-deleted Ror2 to native Ror2.

The in-vivo FCCS technology is state-of-the-art technique and the requested experiments are extremely time and labour intensive. Therefore, we would rather tackle the experimental suggestions in an alternative way: To improve our data set, we provided an extended analysis of co-localization with the CRD deleted Ror2 (Figure 2B, Figure 2—figure supplement 1G). We added a Pearson colocalization analysis to demonstrate the significant alterations in co-localization between the individual constructs. We further improved description in the text.

2) The presentation of Wnt8a transport and Lrp6 clustering in Figure 5 and Figure 5—figure supplement 1 is confusing. Additionally the authors describe the formation of Lrp6-GFP clusters but it is not clear if clustering dynamics have been observed or only fully formed clusters.

We improved the presentation of the data in Figure 5. Furthermore, we added the description about the experimental settings (e.g. focal plane) and added orthogonal views if possible. Furthermore, we changed the text accordingly and cited Hagemann et al. describing cluster dynamics (e.g. formation, endocytosis, and degradation) in zebrafish (subsection “Ror2 presents Wnt8a to the target cell to induce ligand-receptor cluster”).

Delineation of the target cell from brightfield comparison does not seem to match evident cell boundaries – is this a consequence of a different focal plane? The statement that of clusters "dissolve" in the membrane of the extending cell is unclear ("enter," "transit"?).

We re-analysed the 3D data set and improved and re-adjusted cell boundaries (Figure 5A, Figure 5—figure supplement 1B). We further changed the description accordingly (subsection “Ror2 induces Wnt-cytonemes during zebrafish neural patterning”).

Finally, to better demonstrate that Wnt8 is actually endocytosed it would seem useful to display orthogonal views to illustrate the presence of reporter within the target cell.

We included orthogonal views in Figure 5E to illustrate cluster co-localization from different angles. Figure 5B, C, F display only single confocal planes. For this analysis the optimal microscope settings (e.g. pinhole) was chosen and described in the text.

3) Statistical analyses need to be improved. Besides the issue raised in #1 above, the authors should throughout provide test statistics with degrees of freedom (and thus indication of sample size) in addition to P values. Presentations of significant differences for means should also include overall ANOVA with corrections for multiple comparisons.

We agree with the reviewer that a multiple parameter comparison by ANOVA would improve the value of the manuscript. Therefore, we added ANOVA tests to the following Figures: 2E, 3D, 4B, D, 6D, 7C, D, F, G. We also included the individual p values into the figure legends. In addition, we added the sample size to the following experiments: Figure 3D, 6D, 7C, E, F, D, G.

Since experimental treatments seem to be sometimes compared to controls and sometimes to one another, it would be preferable to simply indicate means that are not significantly different from one another in post hoc analyses (e.g., using letter codes).

We improved the presentation of the quantifications accordingly. In exceptional cases we calculated and displayed significant differences between individual experiments (e.g. constructs). These were highlighted specifically to avoid confusion.

It is also unclear whether data obtained meet the assumptions of the statistics used. For example, in the second paragraph of the subsection “Ror2 dependent cytonemes in cancer cells and intestinal organoids”, it states that IRSp53-4K/Ror2 cells are significantly reduced in proliferation compared to Ror2 cells. Yet a major difference illustrated in the plot would seem to be a change in variance, which would likely require a transformation for analyzing appropriately, and it is not clear from the plot what the significance level of this comparison might be.

We added the t-test value >0.05 (NS) to Figure 7 and we changed the subsection “Ror2-dependent cytonemes in cancer cell lines and intestinal organoids”. In addition, we described the observed change in variance.

Similarly in the first paragraph of the subsection “Ror2 induces Wnt-cytonemes during zebrafish neural patterning”, it notes that significant differences in filopodia numbers and lengths were observed across stages, yet Figure 4B does not clearly illustrate stage differences in lengths, only numbers.

We changed this in the text accordingly (subsection “Ror2 induces Wnt-cytonemes during zebrafish neural patterning”, first paragraph).

Finally, it is not clear why the authors present individual data points in 7G but only histograms or box plots elsewhere.

We changed the presentation of the data in Figure 7G to a boxplot to match the presentation in the other figures.

4) The ways in which authors distinguish cytonemes from other filopodia are not clear.

We added a definition of cytonemes and clarified this in the text (subsection “Ror2 induces Wnt-cytonemes during zebrafish neural patterning”).

The data presented in Figure 4F to argue that are Ror2 regulates a specific class of Wnt8a+ cytonemes also need clarification. For instance, are the number of Wnt8+ and Wnt8- cytonemes (filopodia?) being compared within cells, or totals across cells?

Text has been changed accordingly (subsection “Ror2 induces Wnt-cytonemes during zebrafish neural patterning”).

Given the alternative "states" of cytonemes, comparisons involving chi-square analyses would seem more appropriate. Likewise the example cells illustrated in Figure 4E seem to be especially extreme cases, and so perhaps not representative of the overall dataset.

We split Figure 4F into two bar diagrams showing filopodia and cytonemes for better visualization. Furthermore, we improved the statistical analysis in Figure 4D. We also added a stacked cytoneme diagram with Chi-square analysis as requested (Figure 4—figure supplement 1L).

5) The simulation results in Figure 4G are potentially interesting as a follow up to the prior work. However, the authors need to more clearly indicate how the changes in numbers of cytonemes used in the simulations correspond to those obtained empirically: are they over the same order of magnitude? There also should be more explanation as to whether the simulations account for variation in lengths of individual cytonemes, and changes in the frequencies of cytonemes across stages as observed in Figure 4B. Finally, the simulations would be more powerful were it possible to assign specific times (e.g., hpf) or distances (μM), rather than relative values only.

Based on the suggestions of the reviewer, we improved Figure 4G and improved the description of the data in the Results section and in the corresponding figure legend. Furthermore, in the Materials and methods section, we have described the modeling system in more detail and explain the parameters for the different simulations. In particular, the number, length and angle distribution of cytonemes are set to the experimentally obtained numbers. As requested, we also included specific times and absolute distances into the model.

6) The authors have several useful reagents for manipulating cytonemes. But there is always a concern about off-target effects of dominant negative constructs, and the potential that other pathways could be involved. With regard to Ror2 in particular, it would be useful to know whether the Ror2 morpholino used for analyses of PCP signaling, causes a loss of cytonemes similar to that of the dominant negative.

We agree with the reviewer on the need of data supporting the function of Ror2 by a specific knock-down. Therefore, we analysed the formation of cytonemes in Ror2 deficient cells after Morpholino-based knock-down. The observed phenotype supports our findings from the studies with the DN-Ror2. We added quantifications of filopodia and Wnt8a-GFP cytoneme with Ror2 MO to Figure 4E and F (In split bar diagrams, see our second response comment 4). Furthermore, we added a t-test analysis to confirm significance.

Likewise, are cell behaviors unrelated to cytoneme extension and signaling unchanged in cells manipulated for Ror2 function, IRSp53, etc.?

We believe that cell behaviour is unchanged as we do not observe an obvious difference in e.g. migration behaviour of Ror2-KD cells compared to Ctrl cells.

Indeed the evidence that cytoneme formation is strictly dependent n PCP signaling is somewhat circumstantial in the sense that defects in convergent extension are consistent with PCP defects, and Ror2 is known to function in PCP signaling; the authors might therefore be more circumspect about potential roles for other pathways or other ways in which Ror2 might function.

We specified the description of Ror2 and indicated that – besides other functions – Ror2 regulated cytonemes (subsection “Ror2 induces Wnt-cytonemes during zebrafish neural patterning”). The reviewer is correct in stating that our study does not rule out the influence of further signalling pathways regulating cytoneme emergence. Therefore, we re-phrased the text and added this possibility in the Discussion section.

7) The authors describe a computational model that supports their experimental observation. While this adds to strength of the paper, I felt not enough detail about this model was provided for me to evaluate it.

We provided more detail to the simulation in particular in the Materials and methods section. For more details see also response to point 5.

8) Figure 2 presents data to demonstrate the specificity of interaction between Wnt8a-GFP and its receptor, Ror2-mcherry. With the data in Figure 2A/B, they attempt to show that these proteins interact, based on the colocalization of these proteins at the cell surface. Although they show an example of a correlative intensity tracing (Figure 2B), it is unfortunately not evident for this to be the case in the raw data shown in Figure 2A/middle panel: the fluorescence intensity of Ror2-mCherry appears uniform along the plasma membrane particularly at the sites showing the Wnt8-GFP spots. I think it is important to show not only more examples, but also to present a serious statistical analysis of how often and to what extent they detect colocalization.

We improved the presentation of the data set and added a Pearson’s Colocalization analysis to quantify our observation. For more details see also our response to point 1.

Figure 2C/E shows an example of IsFCS analysis. This example is consistent with molecular association in regions of colocalization based on similar cross correlation and auto-correlation profiles and longer lag times, hence slower diffusion. As indicated above, I think it is important to show more examples, and clearly describe the number of events that where analyzed and clarify whether it is always the case that colocalization equates to complex formation.

We improved our description (subsection “Cluster formation of Wnt8a and Ror2 is dependent on the CRD domain”) and added technical details to the analysis. For further details see also our second response to point 1.

Not clear from this analysis, is the possible influence that expression level might have on the outcomes and determinations. I think this is important particularly since it is likely that the tagged proteins are expressed at a significantly higher level than endogenous. The authors should also indicate the sensitivity of their microscopes, so that one can estimate the number of proteins that engaged as complexes.

To rule out over expression artefacts of the tagged constructs, we expressed Ror2-mCherry at a sub-functional level. For further information see also our response to the Editor’s comment.

This is technical difficult and could lead to a wrong interpretation. Therefore, we would like to avoid to estimate the number of proteins that engaged as complex.

[Editors' note: further revisions were requested prior to acceptance, as described below.]

Thank you for attending to issues that were previously raised, but we are not convinced that the revised manuscript addresses all of the substantive concerns. I would think that the following issues could be addressed without undue further effort or delay.1) The robustness of colocalization analyses remain unclear given the apparently limited sampling. Although Pearson coefficients, as well as a different look up table, are helpful, they do not really address the issue of sampling adequacy. With conclusions based apparently on one cell per treatment it is difficult to know how representative these observations might be. Is it possible to expand the analysis to multiple/more cells?

We have added a quantification of the co-localisation measurements to Figure 2. Measuring the fluorescent intensity across (orthogonal) to the membrane allowed us to analyse 10 Wnt8a-Ror2 clusters in 5 different cells from 2 independent embryos per treatment. This analysis supports our hypothesis that Wnt8a co-localizes with Ror2 in the membrane and that the CRD domain facilitates this interaction (see new Figure 2C).

2) Although sample sizes and overall P values have been added in places, the authors still do not report test statistics (e.g., F values) or their associated degrees of freedom, which are also important for evaluating the robustness of their inferences. Similarly they have not addressed whether their data meet the assumptions of ANOVA or t-tests, specifically that variances are similar across treatments and residuals are normally distributed.

We have added the requested the information regarding the ANOVA tests (i.e. F values, df) in the corresponding figure legends and we expanded the Materials and methods section.

3) The issue of whether or not fluorescent proteins are properly localized remains somewhat murky. The authors have provided additional data to show that concentrations used did not result in overt morphological defects. Yet apparently normal morphology does not necessarily imply correct subcellular localization. The analyses would have inspired more confidence if localization had been examined at very low concentrations of injected construct, or across a range of concentrations. Minimally, clear statements of caveats associated with these analyses would seem important.

We have added a statement regarding the localization of the over-expressed, fluorescent constructs and added several references supporting our hypothesis by showing the subcellular localisation of the endogenous proteins in the Discussion.